



# The soil organic carbon stabilization potential of old and new wheat cultivars: a $^{13}CO_2$ labelling study

Marijn Van de Broek[1*], Shiva Ghiasi[2*], Charlotte Decock[1,3], Andreas Hund[4], Samuel Abiven[5], Cordula Friedli[4,5], Roland A. Werner[2], Johan Six[1]

[1]Sustainable Agroecosystems group, Department of Environmental Systems Science, Swiss Federal Institute of Technology, ETH Zürich, Zürich, Switzerland
[2]Grassland Sciences group, Department of Environmental Systems Science, Swiss Federal Institute of Technology, ETH Zürich, Zürich, Switzerland
[3]California State University, San Luis Obispo, CA, USA
[4]Group of Crop Science, Department of Environmental Systems Science, Swiss Federal Institute of Technology, ETH Zürich, Zürich, Switzerland
[5]Department of Geography, University of Zürich, Zürich, Switzerland
*Marijn Van de Broek and Shiva Ghiasi contributed equally to this study

*Correspondence to*: Marijn Van de Broek (Marijn.vandebroek@usys.ethz.ch)

**Abstract.** Over the past decades, average global wheat yields have increased by about 250 %, mainly due to the cultivation of high-yielding wheat cultivars. This selection process not only affected aboveground parts of plants, but in some cases also reduced the root biomass, with potentially large consequences for the amount of organic carbon (OC) transferred to the soil. To study the effect of wheat breeding for high-yielding cultivars on subsoil OC dynamics, two old and two new wheat cultivars from the Swiss wheat breeding program were grown for one growing season in 1.5 m-deep lysimeters and pulse-labelled with $^{13}CO_2$, to quantify the amount of assimilated carbon that was transferred belowground and potentially stabilized in the soil. The results show that although the old wheat cultivars with higher root biomass transferred more assimilated carbon belowground compared to more recent cultivars, no significant differences in net soil organic carbon (SOC) stabilization were found between the different cultivars. As a consequence, the long-term effect of wheat cultivar selection on SOC stocks will depend on the amount of root biomass that is stabilized in the soil. Our results suggest that the process of wheat selection for high-yielding cultivars resulted in lower amounts of belowground carbon translocation, with potentially important effects on SOC stocks. Further research is necessary to quantify the long-term importance of this effect.

## 1 Introduction

Soil management has a large influence on the size of the soil organic carbon (SOC) stock in managed arable soils. This is evident from the large decrease in SOC that is generally observed after soils under natural vegetation are converted to arable





land (Don et al., 2011; Guo and Gifford, 2002; Poeplau et al., 2011). As a consequence, the mineralization of SOC and the loss of forest caused by land use change has contributed about 30 % to the increase in atmospheric $CO_2$ concentration since

the onset of the industrial revolution (Le Quéré et al., 2018). Current contributions of the agricultural sector to global warming have been estimated to be about 11 %, but are mostly in the form of $N_2O$ and $CH_4$ and not anymore as $CO_2$ (Tubiello et al., 2015).

The rising awareness that there is potentially an opportunity to increase subsoil organic carbon (OC) stocks (Chen et al., 2018) has led to the proposal that agricultural soils can be a sink of atmospheric $CO_2$ by applying appropriate climate-smart

agricultural practices (Chenu et al., 2018; Minasny et al., 2017; Paustian et al., 2016). Multiple management practices have been shown to increase the OC content of cultivated soils, including the application of organic amendments to soils (Sandén et al., 2018), increasing the amount of crop residues returned to the field (Lehtinen et al., 2014) and planting of cover crops (Kong and Six, 2010; Poeplau and Don, 2015). These mechanisms have been studied intensively over the past decades, with multiple reviews suggesting that these management practices have the potential to increase the SOC content of arable soils

(Paustian et al., 1997, 2016; Sandén et al., 2018). In addition, growing crops with deeper roots and/or higher root biomass has been proposed to increase OC sequestration in arable soils (Kell, 2011).

Deep rooting is also proposed by breeders to decrease the effect of drought in climates where deep soil water is available during the main cropping season (Wasson et al., 2012). Increased rooting depth is also proposed as a strategy to mitigate the effects of climate change in temperate climates as exemplified for the Swiss plateau (Friedli et al., 2019). Yet, for improving

yield and resource uptake, the proposed root ideotype is steep, cheap and deep (Lynch, 2013) with less biomass and branching in the upper part of the soil (Wasson et al., 2014). However, a direct or marker-assisted selection for root traits is very rare in conventional breeding programs. Accordingly, we have very limited knowledge if and how breeders alter the root system and potentially affect belowground carbon cycling. One way to evaluate the effect of breeder's selection on root characteristics and subsoil carbon cycling is to compare old and new varieties of the same breeding programme. For the Swiss wheat breeding

programmes, the selection process reduced the mass and depth of roots under well-watered conditions but modern genotypes enhanced root allocation to deep soil layers under drought (Friedli et al., 2019). The negative trend of rooting depth was also present in other breeding programmes (Aziz et al., 2017), but has not been observed consistently (Cholick et al., 1977; Feil, 1992; Lupton et al., 1974). While our understanding of the indirect effects of breeding on root morphology and architecture is limited, to the best of our knowledge, there is no information about the effect of breeding on changes in subsoil OC dynamics

and root respiration.

One reason for the lack of quantitative data about the effects of rooting depth on SOC sequestration is related to difficulties in measuring the amount of carbon transferred from roots to the soil (gross rhizodeposition) and the proportion of carbon that is eventually stabilized there (net rhizodeposition). The fact that rhizodeposition occurs below the soil surface greatly prevents direct observations of this '*hidden half of the hidden half*' of the SOC cycle (Pausch and Kuzyakov, 2018). First of all, direct

measurements of root exudation rates are hampered by the fact that rhizodeposits are used by rhizosphere microorganisms within a couple of hours after they are released, resulting in very low concentrations of root carbon exudates in the soil



(Kuzyakov, 2006). Second, the release of carbon exudates by agricultural crops is not equally divided throughout the growing season, but mainly occurs in the first 1 – 2 months of the growing period and decreases sharply thereafter (Gregory and Atwell, 1991; Keith et al., 1986; Kuzyakov and Domanski, 2000; Pausch and Kuzyakov, 2018; Swinnen et al., 1994). Third,
measurements of the effects of rhizodeposits on changes in SOC stocks are further complicated by the priming effect, i.e. their positive effect on the mineralization of native SOC (Fontaine et al., 2007; de Graaff et al., 2009).

To overcome these difficulties, rates of C rhizodeposition can be measured by labelling the plants with $^{13}CO_2$ or $^{14}CO_2$ (Jones et al., 2009; Kuzyakov and Domanski, 2000) and subsequently tracing the amount of photosynthetically assimilated $^{13}C$ or $^{14}C$ label in the soil at the end of the growing season (Kong and Six, 2010, 2012). Practical limitations, however, complicate the
continuous application of $^{13}CO_2$ or $^{14}CO_2$ during the course of an entire growing season. Therefore, plants are commonly labelled at fixed time intervals during the growing season (repeated pulse-labelling). This results in reliable estimates of the partitioning of assimilated carbon to different plant compartments, as well as into the soil (Kong and Six, 2010; Kuzyakov and Domanski, 2000; Sun et al., 2018).

In addition, assessing the magnitude of the carbon transfer from roots to the soil is not straightforward, particularly under field
conditions. While carbon inputs from crops to the soil are often derived from yield measurements (Keel et al., 2017; Kong et al., 2005; Taghizadeh-Toosi et al., 2016), these quantities are often poorly related to root biomass or the magnitude of root exudates (Hirte et al., 2018; Hu et al., 2018). A better understanding of the factors controlling the rates of carbon rhizodeposition by different agricultural crops is thus necessary to assess how different crops affect SOC cycling and to provide SOC models with reliable rates of carbon inputs to the soil.

The lack of an unambiguous relation between wheat yield and root biomass on the one hand, and sufficient knowledge to reliably convert root biomass to rates of subsoil OC sequestration on the other hand underlines the need for additional research on both issues. This is needed to assess (i) differences in the amount of subsoil OC stabilized by different wheat cultivars and (ii) if breeding for high-yielding wheat cultivars with a lower root biomass has resulted in lower amounts of SOC sequestration. Therefore, the present study used four different bread wheat cultivars from a century of Swiss wheat breeding (Fossati and
Brabant, 2003; Friedli et al., 2019) to test the hypothesis that wheat cultivars with shallow rooting systems and lower root biomass result in less subsoil OC stabilization over the course of a growing season. The experiments were carried out in large mesocosms, which allowed to study the plant-soil system under controlled conditions that closely resemble a field situation.

## 2. Materials and Methods

### 2.1 Experimental set-up

**2.1.1 ETH mesocosm platform**

To assess the effect of wheat root characteristics on subsoil OC stabilization in a realistic soil environment under controlled environmental conditions, an experiment was set up at the mesocosm platform of the Sustainable Agroecosystems Group at





the Research Station for Plant Sciences Lindau (ETH Zürich, Switzerland). The platform was located inside a greenhouse and consisted of 12 cylindrical lysimeters with a diameter of 0.5 m and a height of 1.5 m, constructed using 10 mm wide

polyethylene (Figure S1). The lysimeters were equipped with probes installed at five different depths (0.075, 0.30, 0.60, 0.90 and 1.20 m below the surface) to measure the volumetric moisture content at a temporal resolution of 30 min (ECH$_2$O EC-5, Decagon Devices, US) and to sample soil pore water (Prenart, Frederiksberg, Denmark) and soil pore air (Membrana, Wuppertal, Germany). The lysimeters were filled with mechanically homogenized soil, collected from an agricultural field in Estavayer-Le-Lac, Switzerland. The upper 0.15 m of the lysimeters were filled with topsoil, collected from the A-horizon,

while the remainder (0.15 – 1.35 m depth) was filled with subsoil. The bottom 0.15 m of the lysimeters (1.35 – 1.50 m depth) consisted of a layer of gravel (Blähton, Erik Schweizer, Switzerland), to facilitate drainage of soil water through the bottom of the lysimeters. The top and subsoil had a sandy clay loam texture with 21 % silt, 21 % clay, 58 % sand, and top- and subsoil pH values were 7.8 and 7.5, respectively. The OC concentration of the top- and subsoil was $0.77 \pm 0.01$ % and $0.40 \pm 0.01$ %, respectively, with a C:N ratio of 6.9 and 5.0, respectively. No carbonates were detected in the soil.

At the top of each lysimeter, pneumatically activated chambers were placed, that were automatically closed when applying the $^{13}CO_2$ label (see section 2.1.3). These chambers were made of stainless steel with fitted Plexiglas panes and covered a rectangular area of 0.5 x 0.5 m with an initial height of 0.1 m. Chamber heights were extended with increasing plant height, using one or two height extensions of 0.5 m each (Figure S1).

### 2.1.2 Wheat cultivars and growth conditions

Four wheat (*Triticum aestivum* L.) cultivars from the Swiss wheat breeding program (Fossati and Brabant, 2003; Friedli et al., 2019) with different breeding ages were selected: Mont-Calme 268 (introduced in 1926), Probus (1948), Zinal (2003) and CH Claro (2007). Generally, more recent cultivars of this program on average have more shallow roots and lower root biomass under well-watered conditions compared to the older cultivars (Friedli et al., 2019). CH Claro was selected as a modern variety with relatively deep rooting.

Before the wheat plants were transplanted to the lysimeters, wheat seeds were germinated in a greenhouse for 2 – 3 days on perforated anti-algae foil laid over 2-mm moistened fleece at a warm temperature (20 °C during day and 18 °C during night) and good light conditions. Next, the seedlings were planted in topsoil-filled containers and transferred to a climate chamber for vernalization for 52 days (Baloch et al., 2003). First, the seedlings were kept 45 days at 4 °C, with 8 hours of light per day and a light intensity of 10 kilolux. During the 3 subsequent days, daylight intensity was increased to 36 kilolux, daytime

temperature was increased to 12 °C and night temperature to 10 °C. During the last 4 days, daytime temperature was increased to 16 °C, and night temperature to 12 °C. The relative humidity was maintained at $60 \pm 10$ % during the entire vernalization period. After vernalization, 70 seedlings were transplanted to every lysimeter, corresponding to a plant density of 387 plants m$^{-2}$.

The experimental set-up consisted of a randomized complete block design. Each of the four wheat cultivars was planted in

three lysimeters, i.e. 3 replicates per cultivar, resulting in a total of 12 lysimeters. These were placed in 3 blocks of 4 rows,



where each wheat cultivar was planted in one lysimeter in each block. The plants were grown in the greenhouse for about 5 months, between 24 August 2015 and 1 February 2016. Despite uneven maturing of plants within and between the lysimeters, all plants had reached flowering stage at the time of harvest. Fertilizer was applied to the soil lysimeters a first time on 5 October 2015, at a rate of 84 kg N ha⁻¹, 36 kg $P_2O_5$ ha⁻¹, 48 kg $K_2O$ ha⁻¹ and 9 kg Mg ha⁻¹, and a second time on 4 December

2015, at a rate of 56 kg N ha⁻¹, 24 kg $P_2O_5$ ha⁻¹, 32 kg $K_2O$ ha⁻¹ and 6 kg Mg ha⁻¹. The lysimeters were watered manually twice per week with a similar amount of water, to keep soil moisture close to field capacity. Differences in the amount of water used by the different cultivars resulted in differences in the water content between the cultivars (Figure S2). The temperature in the greenhouse was set to 20 °C during the day and 15 °C during the night. During the experiment, the average temperature in the greenhouse was 16.9 °C, with a minimum and maximum of 9.3 °C and 29.8 °C respectively. The average humidity of 63.7 %,

with a minimum and maximum of 35.3 % and 86.4% respectively.

### 2.1.3 Repeated ¹³C pulse-labelling

In order to study carbon allocation within the atmosphere-plant-soil system, a ¹³C pulse-labelling approach was used. 99% $^{13}CO_2$ (Euriso-top, Saint-Aubin, France) was applied once per week (Thursdays) by injecting 15, 56 or 98 mL $CO_2$ into each chamber depending on the chamber extension used, in order to yield a target $^{13}CO_2$ content of 58%. A weekly labelling

frequency has been shown to ensure a sufficient abundance of root-derived ¹³C in the soil at the end of the experiment (Bromand et al., 2001; Kong and Six, 2010). After chamber closure, $CO_2$ concentration in one chamber was monitored using a $CO_2$ analyzer (Li-820, LICOR, Lincoln, US). After the $CO_2$ concentration dropped below 200 ppm, a $^{13}CO_2$ pulse was injected to yield a post-label $CO_2$ concentration of 570 ppm in the chamber headspace. The chamber lids were kept closed for two hours after label injection to achieve sufficient uptake and then re-opened to avoid condensation. On the same day of

pulse-labelling, all chambers were closed overnight to recuperate ¹³C lost through night respiration and allowed to be taken up by the plants in the morning before reopening the chambers.

### 2.2 Measurements

### 2.2.1 Belowground $CO_2$ concentration and $\delta^{13}CO_2$

Soil gas sampling was performed once per week (Wednesdays) by attaching a pre-evacuated 110 mL crimp serum vial to a

sampling port at each depth, leaving it equilibrating overnight. For each sample, a 20 mL subsample was transferred to a pre-evacuated Labco exetainer (12 mL), and used to determine the $CO_2$ concentration and its carbon isotopic composition ($\delta^{13}C$). The $CO_2$ concentration of each sample was determined using a gas chromatograph equipped with a thermal conductivity detector (Bruker 456-GC, Germany). For a limited amount of samples, the $\delta^{13}C$ value of $CO_2$ was measured with a Gasbench II modified as described by Zeeman et al. (2008) coupled to a Delta^{plus}XP isotope ratio mass spectrometer (IRMS,

ThermoFisher, Germany). The standard deviation of the measurements was < 0.15 ‰.





### 2.2.2 Sampling and general soil analyses

At the end of the experiment, the aboveground biomass of the wheat plants was harvested separately for each lysimeter and separated into leaves, ears and stems. Soil from the lysimeters was collected by destructive sampling to analyze bulk density, root biomass and other soil properties. The sampling was done layer by layer. After a soil layer had been sampled, it was

removed completely from the lysimeter and the next layer was sampled. From each depth increment (0 – 0.15, 0.15 – 0.45, 0.45 – 0.75, 0.75 – 1.05, 1.05 – 1.35 m depth), five soil cores were collected per lysimeter using a soil core sampler (5.08 cm diameter, Giddings Machine Company Inc., Windsor, CO, US). Three of the five cores per lysimeter and depth increment were used for the determination of root biomass based on a combination of buoyancy and sieving through a 530 µm sieve, using a custom-built root washing station. The remaining two soil cores were sieved at 8 mm, air-dried and stored for further analysis.

Prior to air drying, the fresh weight and volume for each core was determined, and a subsample was taken for the determination of gravimetric soil moisture content. Bulk density was calculated based on fresh weight, gravimetric moisture content, and core volume. Soil texture was measured using a particle size analyzer (LS 13 320, Beckman Coulter, Indianapolis, USA). Prior to analysis 0.1g of soil was shaken for 4 h with 4 ml of 10 % Na-hexametaphosphate and sonicated for 1 min.

### 2.2.3 Soil microbial biomass

Soil microbial biomass was extracted from soil samples that had been frozen at -20 °C for 6 months immediately after sampling. Two subsamples of 40 mg were taken from each sieved soil sample. One set was fumigated for 24 hours using chloroform. Next, total dissolved OC was extracted from each fumigated and non-fumigated subsample by shaking it in 200 mL 0.05 M $K_2SO_4$ for one hour, prior to filtering through a Whatman 42 filter paper. Total OC concentrations in $K_2SO_4$ extracts were determined using a CN analyzer (multi N/C 2100 S analyser, Analytik Jena, Germany). The gravimetric soil water content was

determined by drying about 10 g of each soil sample at 105 °C and subtracting the weights before and after drying. The carbon content of the soil microbial biomass was calculated according to Vance et al. (1987) as:

$$TOC_{MB} = \frac{TOC_F - TOC_{NF}}{0.45} \qquad \text{(Eq. 1)}$$

Where $TOC_F$ and $TOC_{NF}$ are the total OC in fumigated and non-fumigated samples, respectively. The remainder of the filtered samples was freeze dried in order to analyze the $\delta^{13}C$ value. The $\delta^{13}C$ value of soil microbial biomass was calculated using mass balance according to Ruehr et al. (2009):

$$\delta^{13}C_{MB} = \frac{(\delta^{13}C_F \cdot C_F - \delta^{13}C_{NF} \cdot C_{NF})}{C_F - C_{NF}} \qquad \text{(Eq. 2)}$$


Where $C_F$ and $C_{NF}$ represent total carbon content of the fumigated and non-fumigated samples, respectively.



### 2.2.4 Organic carbon concentration and isotopic composition of plant material, soil organic carbon and microbes

The OC concentration and isotopic composition ($\delta^{13}$C) of above- and belowground plant material, microbial biomass and soil were measured by weighing 2, 4, 80 and 100 mg, respectively, of each sample into Sn capsules (9 x 5 mm, Saentis, CH) for

analysis with a Flash EA 1112 Series elemental analyzer (ThermoFisher, Germany) coupled to a Delta[plus] XP IRMS via a ConFlo III (Brooks et al., 2003; Werner et al., 1999; Werner and Brand, 2001). The measurement precision (SD) of the quality control standards (tyrosine Tyr-Z1, caffeine Caf-Z1), was 0.37 (‰) for above and belowground plant material, microbes and the soil samples.

### 2.3 Data processing

### 2.3.1 Excess $^{13}$C calculations

The mass of $^{13}$C label that was recovered in (i) the aboveground vegetation, (ii) roots of wheat plants and (iii) the soil was calculated following Studer et al. (2014):

$$m^E\left(^{13}C\right) = \frac{\chi^E\left(^{13}C\right) \cdot m(C) \cdot M\left(^{13}C\right)}{\chi\left(^{12}C\right) \cdot M\left(^{12}C\right) + \chi\left(^{13}C\right) \cdot M\left(^{13}C\right)} \qquad \text{(Eq. 3)}$$


Where $m^E\left(^{13}C\right)$ is the mass of recovered $^{13}$C label (g m$^{-2}$), $\chi^E\left(^{13}C\right)$ is the excess atom fraction (unitless, calculated following Coplen (2011)), $m(C)$ is the total mass (g m$^{-2}$) of C, $M\left(^{12}C\right)$ and $M\left(^{13}C\right)$ are the molar weight of $^{12}$C and $^{13}$C (g mol$^{-1}$), respectively, and $\chi\left(^{12}C\right)$ and $\chi\left(^{13}C\right)$ are the $^{12}$C and $^{13}$C atom fraction (unitless), respectively.

To calculate the excess atom fraction ($\chi^E\left(^{13}C\right)$) of the soil compartment, the isotopic composition of the soil at the start of the

experiment was used as the reference value (-26.45 $\pm$ 0.04 ‰ for the topsoil, -25.01 $\pm$ 0.13 ‰ for the subsoil). As all lysimeters were labelled with $^{13}$CO$_2$, no control treatment for the wheat plants was present. Therefore, a $\delta^{13}$C reference value of -28 ‰ was assumed for the roots of all wheat plants. The calculation of excess $^{13}$C is very sensitive to variability in input parameter values, including the $\delta^{13}$C value of roots and soil. Therefore, a sensitivity analysis was used to show that varying the initial $\delta^{13}$C value of the wheat plants with +/- 3 ‰, a typical range over which $\delta^{13}$C values can vary in the field because of e.g.

precipitation (Kohn, 2010), led to changes in calculated m$^E$($^{13}$C) in the order of +/- 1 % for aboveground biomass and +/- 1 – 5 % for belowground biomass. The effect of the initial $\delta^{13}$C value of the biomass on the calculated amount of recovered $^{13}$C label in the wheat plants was thus limited. In addition, there was a large variability between the replicate lysimeters of the same cultivar for some of the input variables to calculate excess $^{13}$C. Moreover, there were some missing data on the $\delta^{13}$C value of root biomass for a limited number of depth intervals, due to low recovery of root biomass. Therefore, calculations of excess

$^{13}$C were performed for each wheat cultivar by combining the average values for the replicates, precluding the application of an analysis of variance to test treatment effects on excess $^{13}$C.





### 2.3.2 Carbon rhizodeposition

The absolute amount of carbon rhizodeposition for the different depth segments in the lysimeters was calculated following Janzen and Bruinsma (1989):


$$Rhizodeposition\ C = \frac{\chi^{E}(^{13}C)_{soil}}{\chi^{E}(^{13}C)_{root}} \cdot C_{soil} \qquad \text{(Eq. 4)}$$

Where rhizodeposition C is expressed in g kg$^{-1}$ for the considered layer, $\chi^{E}(^{13}C)_{soil}$ and $\chi^{E}(^{13}C)_{root}$ are the excess $^{13}$C atom fraction in the soil and roots respectively, calculated as described in section 2.3.1, and $C_{soil}$ is the OC concentration of the

considered soil layer (g kg$^{-1}$). This approach assumes that the isotopic enrichment of rhizodeposits and roots are equal. The absolute amount of carbon rhizodeposition for each soil layer was calculated by multiplying rhizodeposition C (g kg$^{-1}$) with the carbon content (kg) present in each of the respective layers.

### 2.3.3 Subsoil CO$_2$ production

Depth profiles of subsoil CO$_2$ production in the lysimeters were calculated using the weekly measured depth profiles of CO$_2$

concentration throughout the experiment. To assess the variability among the different lysimeters, these calculations were performed separately for every lysimeter and average CO$_2$ production depth profiles were calculated for each cultivar. Measurements of CO$_2$ concentration, soil temperature and soil moisture content were performed at discrete depths (0.075, 0.30, 0.60, 0.90 and 1.20 m depth). Continuous depth profiles of these variables at a vertical resolution of 0.05 m were obtained using linear interpolation. Depth profiles of CO$_2$ production were calculated using the discretized form of the mass balance

equation of CO$_2$ in a diffusive one-dimensional medium, following Goffin et al. (2014):

$$P(z)_i = \frac{\Delta(\varepsilon_i[CO_2]_i)}{\Delta t} + \frac{F_{top_i} - F_{bot_i}}{\Delta z} \qquad \text{(Eq. 5)}$$

Where $P(z)$ is the CO$_2$ production in layer $i$ (µmol CO$_2$ m$^{-3}$ s$^{-1}$) over timespan $\Delta t$, $t$ is the time (s), $\varepsilon_i$ is the air-filled porosity

in layer $i$ (m$^3$ m$^{-3}$), $[CO_2]_i$ is the CO$_2$ concentration of layer $i$ (µmol CO$_2$ m$^{-3}$), $F_{top_i}$ and $F_{bot_i}$ are the CO$_2$ fluxes transported through the upper and lower boundaries of layer $i$ (µmol CO$_2$ m$^{-2}$ s$^{-1}$) during timespan $\Delta t$, respectively, and $z$ is the depth (m). The vertical CO$_2$ fluxes are calculated as (Goffin et al., 2014):

$$F_{top_i} = -\overline{D}_{si-1,i} \frac{[CO_2]_{i-1} - [CO_2]_i}{\Delta z} \qquad \text{(Eq. 6)}$$

$$F_{bot_i} = -\overline{D}_{si,i+1} \frac{[CO_2]_i - [CO_2]_{i+1}}{\Delta z} \qquad \text{(Eq. 7)}$$





Where $\bar{D}_{si,j}$ is the harmonic average of the effective diffusivity coefficient ($D_s$) between layers $i$ and $j$, and $\Delta z$ is the layer thickness. The effective diffusivity coefficient is calculated using a formula appropriate for repacked soils (Moldrup et al., 2000):


$$D_{s,i} = D_{0,t} \frac{\varepsilon_{i,t}^{2.5}}{\Phi_i} \qquad \text{(Eq. 8)}$$

Where $D_0$ is the gas diffusion coefficient of $CO_2$ in free air over timespan $\Delta t$ (m$^2$ s$^{-1}$), $\varepsilon_i$ is the air-filled porosity of layer $i$ over timespan $\Delta t$ (m$^3$ m$^{-3}$) and $\Phi_i$ is the total soil porosity of layer $i$ (m$^3$ m$^{-3}$). The total soil porosity was calculated as $\Phi_i = 1 - \rho_i/\rho_p$

where $\rho_i$ is the soil bulk density (ton m$^{-3}$) and $\rho_p$ is the particle density (2.65 ton m$^{-3}$). Due to the large vertical variability in measured bulk density depth profiles, a constant bulk density profile was assumed for the subsoil (below 0.15 m depth), calculated as the average of the measured bulk density values for these layers. The air-filled porosity over timespan $\Delta t$ was calculated as the difference between the total porosity (m$^3$ m$^{-3}$) and the average measured water-filled pore space over timespan $\Delta t$ (m$^3$ m$^{-3}$). The latter was measured throughout the experiment (see section 2.1.1) and corrected based on differences between

these measurements at the end of the experiment and the measured volumetric water content of the sampled soil at the end of the experiment. For this purpose, different correction equations were used for (i) the upper soil layer (0 – 15 cm) and (ii) all deeper layers combined.

The gas diffusion coefficient in free air was corrected for the individual lysimeters for variations in temperature and soil moisture throughout the experiment (Massman, 1998), as:


$$D_0 = D_{0,stp} \frac{p_0}{p} \left(\frac{T}{T_0}\right)^\alpha \qquad \text{(Eq. 9)}$$

Where $D_{0,stp}$ is the gas diffusion coefficient for $CO_2$ in free air under standard temperature (0 °C) and pressure (1 atm) (1.385 · 10$^{-5}$ m$^2$ s$^{-1}$ (Massman, 1998)) and $\alpha$ is a coefficient (1.81; Massman, 1998)). Semi-continuous measurements of soil

temperature in every lysimeter were used to calculate $D_0$ values throughout the experiment, while a constant atmospheric pressure of 1 atm throughout the experiment was assumed.

To obtain depth profiles of the total amount of $CO_2$ produced by the different wheat cultivars during the experiment (expressed as g $CO_2$ m$^{-2}$), the calculated $CO_2$ production rates between all measurement days ($P(z)$) were summed for the timespan of the experiment and converted to g $CO_2$ m$^{-2}$ using the molecular mass of $CO_2$ (44.01 g mol$^{-1}$). We applied the boundary condition

of the absence of a flux of $CO_2$ at the bottom of the lysimeters. It is noted that these calculations do not make a distinction between the source of $CO_2$ of production, thereby combining both autotrophic and heterotrophic $CO_2$ production (total soil respiration). For more information about these methods, reference is made to Goffin et al. (2014).



**2.4 Statistics**

To account for the three blocks in the randomized complete block design, statistically significant differences between
characteristics of different cultivars (summed for the different depth layers) were checked using a two-way analysis of variance
(anova) without interactions followed by a Tukey test, based on the values obtained for the individual replicates (n = 3 for
every cultivar). This was done after checking for homogeneity of variance (Levene's test) and normality (Shapiro-Wilk test)
using a confidence level of 0.05. The effect of cultivar and depth on soil bulk density and belowground biomass was assessed
using a three-way anova, with cultivar and depth being fixed effects and blocks being treated as a random effect. In addition,
a pair-wise comparison of the results of the three-way anova were used to check for statistically significant differences in
belowground biomass and bulk density between the cultivars, when accounting for the effect of depth. Belowground biomass
was log-transformed to increase normality and homogeneity of variances for the latter analysis. Statistical analyses were
performed in Matlab®. Uncertainties on reported variables are expressed as standard errors (n = 3).

**3. Results**

**3.1 Aboveground biomass**

The aboveground biomass produced at the end of the experiment was significantly different between Zinal ($710 \pm 114$ g m$^{-2}$)
and Probus ($1154 \pm 220$ g m$^{-2}$), while the aboveground biomass of CH Claro ($1064 \pm 207$ g m$^{-2}$) and Mont-Calme 268 ($1119 \pm 174$ g m$^{-2}$) was not significantly different from any other cultivar (Figure 1, Table 1). The biomass of the ears was
significantly higher for Zinal ($333 \pm 68$ g m$^{-2}$), compared to CH Claro ($92 \pm 33$ g m$^{-2}$), Probus ($21 \pm 12$ g m$^{-2}$) and Mont-Calme
($13 \pm 8$ g m$^{-2}$) (Figure 1, Table S1). It is noted that these data should be interpreted with care, since not all plants reached
maturity at the time of harvest, and is potentially not representative for the biomass of the ears of full-grown plants. No
significant differences were found between the $\delta^{13}$C values of aboveground biomass of the different cultivars (Figure 2). The
high $\delta^{13}$C values of the aboveground biomass of all wheat cultivars (266 ‰ on average) showed that a substantial amount of
the $^{13}$CO$_2$ tracer was incorporated by all wheat plants (Figure 2).

**3.2 Belowground biomass**

The average root biomass was highest in the topsoil and significantly lower in the subsoil layers of all four wheat cultivars
(Figure 1B). Significant differences between the root biomass of the different cultivars were analyzed (i) using total root
biomass as the dependent variable (using a two-way anova without interactions, with cultivar and block as fixed effects) and
(ii) using the root biomass per depth layer (using a three-way anova, with cultivar and depth as fixed effects, and block as
random effect, after log-transformation of the data). When using total root biomass summed for the different depths, the
average total root biomass was not significantly different between the cultivars, but was on average largest for the older wheat
cultivars ($205 \pm 67$ g m$^{-2}$ for Mont-Calme 268 and $161 \pm 54$ g m$^{-2}$ for Probus) and lowest for the more recent cultivars ($97 \pm$





20 g m$^{-2}$ for Zinal and 107 $\pm$ 28 g m$^{-2}$ for CH Claro) (Table 1). In contrast, when depth was included as an independent variable, the root biomass of Zinal was significantly lower compared to the root biomass of Probus and Mont Calme 268, while the root

biomass of CH Claro was not significantly different from any of the other cultivars (Figure1B). These differences were mostly present in the two uppermost soil layers, while root biomass was not significantly different between different cultivars at any depth, except for Zinal and Mont-Calme 268 between 0.45 – 0.75 m depth (Figure 1). The root:shoot ratio varied between 0.10 $\pm$ 0.02 and 0.19 $\pm$ 0.08, and was not significantly different between the different cultivars (Table 1).

The depth profiles of the $\delta^{13}$C of root biomass were different between the old and more recent wheat cultivars (Figure 2). In

the two uppermost soil layers, no significant differences were detected between the $\delta^{13}$C values of root biomass of the different cultivars. These differences could not be checked for statistically significant differences in deeper soil layers due to a lack of sufficient recovered root biomass in each block. The $\delta^{13}$C values of the roots of the old wheat cultivars showed only limited variation with depth, with values between ca. 150 and 200 ‰. In contrast, the $\delta^{13}$C values of the roots of the more recent wheat cultivars were highest in the two uppermost soil layers (0 – 45 cm) and showed an abrupt decrease with depth in deeper soil

layers. Older wheat cultivars thus allocated more $^{13}$C label to their roots compared to the more recent cultivars.

### 3.3 Soil and soil organic carbon characteristics

The SOC concentration in the lysimeters was similar to the OC concentration of the initial soil (Figure 3A). A direct comparison between the SOC concentration before and after the experiment could not be made, as no measurements of the OC concentration of the soil in the lysimeters before the start of the experiment could be made. However, the SOC

concentration measured at the different depths in the lysimeters was similar to the OC concentration measured on the soil that was used to fill the lysimeters (Figure 3A). No statistically significant differences in SOC concentration were found between the different cultivars at any depth.

The SOC in the two uppermost soil layers (0 – 45 cm) of all wheat cultivars was enriched in $^{13}$C compared to the soil that was used to fill the lysimeters (Figure 3B). Although the $\delta^{13}$C value of SOC was not significantly different at any depth between

any of the cultivars, the largest increase in the $\delta^{13}$C value of topsoil OC was observed for Probus and Mont-Calme 268 (Figure 3B), indicating that the soil under the old cultivars incorporated more of the $^{13}$C label, compared to the more recent cultivars. The limited difference between (i) the $\delta^{13}$C values of the soil used to fill the lysimeters and (ii) the measurements at the end of the experiment below a depth of 0.45 m, indicates a lower amount of incorporated $^{13}$C label in the subsoil. Similarly, the $\delta^{13}$C value of topsoil microbial biomass was more positive compared to deeper soil layers for all cultivars, indicating that microbes

utilized more substrate enriched in $^{13}$C in the two uppermost soil layers, compared to deeper soil layers (Figure 3C). Statistically significant differences were only detected in the layer between 0.15 and 0.45 m depth, where the $\delta^{13}$C value of microbial biomass under Zinal was significantly lower compared to Mont-Calme 268. Depth profiles of microbial biomass carbon were relatively constant (200 – 500 µg C g soil$^{-1}$) with no consistent differences between different cultivars (Figure S3). The $\delta^{13}$C values of soil CO$_2$ ($\delta^{13}$CO$_2$) at the end of the experiment were similar for all wheat cultivars for the two uppermost layers (0 –

0.45 m) (Figure 3D). Deeper down the profile, the $\delta^{13}$CO$_2$ under the old wheat cultivars was more enriched in $^{13}$C compared





to the more recent cultivars, by an average of ca. 30 ‰. The only statistically significant differences were detected in the lowermost layer, where the $\delta^{13}C$ value of $CO_2$ of Zinal and CH Claro were significantly lower compared to Mont-Calme 268. There was no significant effect of cultivar on the bulk density of the soil at the end of the experiment ($F_{3,59} = 1.9$, $p = 0.23$), while there was a significant effect of depth on bulk density ($F_{4,59} = 19.4$, $p < 0.0005$). The average bulk density of all lysimeters

was highest in the topsoil ($1.67 \pm 0.12$ Mg m$^{-3}$) and showed substantial variation with depth (Figure S4A). The gravimetric moisture content in the lysimeters at the end of the experiment increased with depth for all cultivars, from ca 0.1 g g$^{-1}$ in the top layer to ca. 0.15 g g$^{-1}$ in the bottom layer (Figure S4B), and was only significantly different between Mont-Calme 268 and Zinal in the uppermost soil layer. The soil moisture content changed relatively little throughout the experiment for all lysimeters, after an initial phase of decreasing soil moisture content at the onset of the experiment (Figure S2).

### 3.4 Excess $^{13}C$ and carbon rhizodeposition

The total amount of $^{13}C$ label that was present in the plant-soil system at the end of the experiment, expressed as excess $^{13}C$, varied substantially between different wheat cultivars (Figure 4A). The lowest amount of $^{13}C$ label was found in the Zinal lysimeters (1.28 g m$^{-2}$), followed by CH Claro (1.64 g m$^{-2}$) and the older wheat cultivars (2.14 g m$^{-2}$ for Mont-Calme 268 and 2.18 g m$^{-2}$ for Probus), with the majority of $^{13}C$ tracer in the above-ground biomass (Figure 4A). Despite these differences, the

relative distribution of the assimilated $^{13}C$ between aboveground biomass, roots and soil was similar between the different wheat cultivars (Figure 4B). On average, $79.8 \pm 9.4$ % of the assimilated tracer ended up in aboveground biomass, $6.6 \pm 1.2$ % in root biomass and $13.6 \pm 2.6$ % in the soil. It is noted that root-respired $^{13}C$ label is not included in this analysis, which may lead to an underestimation of the fraction of $^{13}C$ label that was allocated belowground.

The total amount of rhizodeposition carbon measured at the end of the experiment down to 0.75 m decreased with depth for

all wheat cultivars, with the exception of Zinal (Figure 4C), with the highest values for Probus ($126 \pm 57$ g m$^{-2}$), followed by CH Claro ($112 \pm 39$ g m$^{-2}$), Zinal ($100 \pm 39$ g m$^{-2}$) and Mont-Calme ($85 \pm 27$ g m$^{-2}$). There was thus no relationship between the amount of rhizodeposition C and year of release of the wheat cultivars.

### 3.5 $CO_2$ concentration and production

Throughout the experiment, the change in the $CO_2$ concentration of the two uppermost soil layers was limited, with average

values for the topsoil between 470 and 761 ppm for all cultivars (Figure 5). Deeper down the lysimeters, relatively constant $CO_2$ concentrations were observed during the first 3 weeks of the experiment, ca. 5.000 – 10.000 ppm. After 3 weeks, subsoil $CO_2$ concentrations abruptly increased and remained high throughout the experiment. These were substantially larger for the older cultivars (with maximum values of ca. 30.000 ppm) compared to the younger cultivars (with maximum values ca. 24.000 ppm).

Despite these high $CO_2$ concentrations in the subsoil, $CO_2$ production was mainly taking place in the topsoil, with the highest rates of $CO_2$ production between 0.10 and 0.20 m depth for all cultivars (Figure 6). For the young cultivars (Zinal and CH Claro), 95 % of $CO_2$ was produced above a depth of 0.3 m. 95 % of $CO_2$ was produced above a depth of 0.55 and 0.6 m for





Probus and Mont-Calme 268, respectively. Despite these observations, neither the calculated total amount of subsoil $CO_2$ production or the depth above which 95 % of $CO_2$ was produced were significantly different between any of the cultivars.

**4. Discussion**

**4.1 Plant carbon dynamics and $CO_2$ production**

No consistent differences in total aboveground biomass between old and new wheat cultivars were observed, although the aboveground biomass of Zinal ($710 \pm 114$ g m$^{-2}$) was substantially lower compared to the other wheat cultivars (on average $1112 \pm 116$ g m$^{-2}$). The observed aboveground biomass values were at the high end of reported values for wheat plants in the

field (Mathew et al., 2017), while the lack of consistent differences in the biomass of wheat cultivars released over a time span of multiple decades has generally been observed (Brancourt-Hulmel et al., 2003; Feil, 1992; Lupton et al., 1974; Wacker et al., 2002).

The fraction of biomass in the grain-bearing ears was, however, much larger for the modern wheat cultivars (on average 9 and 47 % of total aboveground biomass for CH Claro and Zinal respectively) compared to the old wheat cultivars (on average 1

and 2 % for Mont-Calme 268 and Probus respectively). While an increase in the fraction of biomass allocated to grains is generally observed when old and modern wheat cultivars are compared (Brancourt-Hulmel et al., 2003; Feil, 1992; Shearman et al., 2005), mostly as a consequence of the introduction of reduced height genes (Tester and Langridge, 2010), the harvest index reported here for the old cultivars might have been slightly underestimated because older cultivars where not yet fully mature at plant harvest.

The total root biomass of the older wheat cultivars was substantially larger compared to the more recent cultivars, although these differences were not statistically significant due to the large variation among different replicates (Table 1). These differences were mostly apparent in the top 0.45 m of the lysimeters (Figure 1). It is not clear if the lack of statistically significant differences in the root biomass within the deeper soil layers was due to (i) inability to collect all fine roots from the soil or (ii) differences in root architecture. These results are in line with a recent study on the biomass of roots of different

wheat cultivars of the Swiss wheat breeding program, including the cultivars used in our experiments (Friedli et al., 2019). This study showed that under well-watered conditions, older wheat cultivars had a substantially higher root biomass compared to the more recently released wheat cultivars. Similar results have been obtained for wheat cultivars released in e.g. Australia (Aziz et al., 2017) and other countries around the world (Waines and Ehdaie, 2007). The root:shoot ratio of the wheat cultivars in our study ($0.10 \pm 0.2 – 0.19 \pm 0.08$, Table 1) were at the low end of reported values for wheat plants globally (Mathew et

al., 2017), but in line with reported values for wheat cultivars of the Swiss wheat breeding program, including the cultivars used in our study (0.14, as measured by Friedli et al. (2019)).

The maximum rooting depth was similar between the old and recent wheat cultivars (Figure 1B). This is in contrast with the results from Friedli et al. (2019), who found that the older wheat cultivars had deeper roots (the depth above which 95 % of roots were found ($D_{95}$) was on average 101 cm) compared to the more recent cultivars included in the present study (average





D$_{95}$ of 85 cm). These differences might partly arise from the different set-up used in both studies. Both experiments were carried out in a controlled greenhouse environment, but Friedli et al. (2019) used soil columns with a diameter of 0.11 m, while in our study lysimeter with a diameter of 0.5 m were used. Additional information about subsoil root dynamics could be obtained from the measured depth profiles of the $CO_2$ concentration and $^{13}CO_2$, with the latter only being measured in the last phase of the experiment. The calculated depth profiles of $CO_2$ production showed that $CO_2$ was being produced down to greater

depths under the old wheat cultivars (Figure 6). Combined with the higher $\delta^{13}C$ values of subsoil $CO_2$ of the lysimeters under the old wheat cultivars at the end of the experiment (Figure 3D), this suggests that the roots of the old wheat cultivars were respiring $CO_2$ down to greater depths compared to the recent wheat cultivars. It is noted that subsoil $CO_2$ was not partitioned between $CO_2$ originating from (i) root respiration and the heterotrophic respiration of root-derived OC by microorganisms and (ii) heterotrophic respiration of native SOC by microorganisms due to lack of data.

In addition, the $\delta^{13}C$ values of root biomass suggest that the temporal root carbon dynamics of the old and recent wheat cultivars differed substantially (Figure 2B). The root biomass of the old wheat cultivars had a high $\delta^{13}C$ value at all measured depths, indicating that the $^{13}CO_2$ label was allocated to the roots at all depths throughout the experiment. In contrast, the root biomass of the recent wheat cultivars was greatly enriched in $^{13}C$ in the top 0.45 m, while deeper roots were much less enriched in $^{13}C$. This suggests that both old and more recent wheat cultivars grew roots down to depths of $> 1$ m in the beginning of the

experiment, while only the old cultivars kept on allocating carbon down to deep roots ($> 0.45$ m) throughout the experiment. The similar $\delta^{13}C$ value of the aboveground biomass of all wheat cultivars (Figure 2A) suggests that the differences in $\delta^{13}C$ values of the root biomass are unlikely to be caused by differences in the relative amount of $^{13}CO_2$ assimilated by the plants, relative to unlabeled $CO_2$. Thus, these results corroborate the hypothesis that old wheat cultivars allocate photosynthates down to their roots throughout a substantial part of the plant growth phase, while this is not the case for more recent cultivars.

**4.2 Carbon allocation by wheat plants**

Subsoils receive the largest portion of OC inputs through dead roots and rhizodeposition (Jones et al., 2009; Nguyen, 2003). To correctly simulate subsoil carbon dynamics, it is therefore important to reliably estimate the fraction of assimilated plant carbon that is transferred to roots and eventually released to the soil. However, estimations of the magnitude of these carbon inputs to the soil are prone to large uncertainties (Oburger and Jones, 2018). A first reason for this is a lack of knowledge on

the root biomass of crops. Although the root:shoot ratio of most common crops is well-known (e.g. Bolinder et al. (2007), Mathew et al. (2017)), it has recently been shown that calculating root biomass based on aboveground plant biomass generally leads to erroneous results (Hirte et al., 2018; Hu et al., 2018; Taghizadeh-Toosi et al., 2016). Secondly, although the partitioning of belowground carbon inputs into root biomass, root respiration and rhizodeposition is relatively constant among different species (if the same growth period is considered (Kuzyakov and Domanski, 2000)), this knowledge is based on a limited

number of available studies (e.g. 20 studies for crops in Pausch and Kuzyakov (2018)). In addition, qualitative and quantitative information about root exudates can be greatly influenced by the methodology used, adding to the uncertainty of available data (Oburger and Jones, 2018). Since uncertainties about the magnitude of carbon inputs to the soil have a great effect on





simulations of SOC dynamics (Keel et al., 2017), there is a great need for additional data collection, both in lab environments as well as in the field.

The partitioning of the $^{13}$C label was very similar between the different wheat cultivars (Figure 4B). It is noted that the amount of rhizosphere-respired $^{13}CO_2$ could not be included in these calculations, although this typically accounts for ca. 7 - 14 % of assimilated carbon in crops, or 40 % of total belowground C allocation (Kuzyakov and Domanski, 2000; Pausch and Kuzyakov, 2018). The fraction of assimilated carbon that is transferred belowground reported here is therefore underestimated. The belowground transfer of ca. 20 % of assimilated C for all cultivars is in line with previous studies, which have reported fractions

of similar magnitude for wheat plants, when not accounting for rhizosphere $CO_2$ respiration: 18 – 25 % (Hirte et al., 2018), 18 % (as reviewed by Kuzyakov and Domanski (2000)), 15 % (Keith et al., 1986), 17 % (Gregory and Atwell, 1991) and 31 % (Sun et al., 2018). In contrast, reported values of the partitioning of belowground translocated carbon by wheat plants to (i) roots and/or (ii) net rhizodeposition are much more variable. The same studies reported net rhizodeposition carbon as a percentage of total belowground carbon (root carbon and net rhizodeposition carbon combined) for wheat plants to be between

23 (as summarized by Kuzyakov and Domanski (2000)) and 72 % (Sun et al., 2018). The results obtained here (68 %) are thus at the high end of reported values. However, they were similar to results from a field study in Switzerland which used two modern Swiss wheat cultivars, among which CH Claro (58 %; Hirte et al. (2018)).

### 4.3 Rates of net carbon rhizodeposition

The total amount of carbon assimilated by the wheat cultivars that was transferred to roots and soil in the top 0.75 m at the end

of the experiment was highest for the older wheat cultivars ($159 \pm 37$ g m$^{-2}$ for Mont-Calme 268, $184 \pm 60$ g m$^{-2}$ for Probus) and lowest for the more recent cultivars ($135 \pm 40$ g m$^{-2}$ for Zinal and $147 \pm 40$ g m$^{-2}$ for CH Claro) (Table 2). It is evident that the total amount of belowground carbon translocation by the wheat plants was higher than these values, as rhizosphere respiration could not be included in our calculations. These numbers are in the range of reported values for wheat plants of 94 – 295 g m$^{-2}$ (as summarized by Keith et al. (1986)), but higher than the value reported by Kuzyakov and Domanski (2000)

(150 g m$^{-2}$), as well as the reported amount by two recent wheat cultivars of the Swiss wheat breeding program (including CH Claro) of 110 – 134 g m$^{-2}$ (Hirte et al., 2018).

In contrast to the total amount of carbon translocated below the soil surface, the amount of net carbon rhizodeposition was not consistently different between the old and more recent wheat cultivars. For the top 0.75 m, this was $85 \pm 27$ g m$^{-2}$ for Mont-Calme 268, $126 \pm 57$ g m$^{-2}$ for Probus, $100 \pm 39$ g m$^{-2}$ for Zinal and $112 \pm 39$ g m$^{-2}$ for CH Claro (Figure 4). These values are

higher compared to values calculated by Pausch and Kuzyakov (2018) (18 – 34 g m$^{-2}$, depth unknown) and Hirte et al. (2018) (63 – 73 g m$^{-2}$; down to 0.75 m depth).

A large uncertainty associated with calculated values of subsoil carbon sequestration using isotopic labelling approaches is related to the assumption that the isotopic enrichment of roots and rhizodeposits are similar (eq. 4). Under this assumption, the $\delta^{13}$C value of roots is used as a proxy for the $\delta^{13}$C of the rhizodeposits. This simplification is made because of the difficulties

in measuring quantitative characteristics of rhizodeposits in a soil medium (Oburger and Jones, 2018), but leads to erroneous





calculations of the amount of carbon rhizodeposition when this assumption is violated (Stevenel et al., 2019). The latter is likely to be the case for two main reasons. First, since a large portion of rhizodeposits is derived from recently assimilated photosynthates (Ekblad and Högberg, 2001; Kuzyakov and Gavrichkova, 2010), the $\delta^{13}C$ values of rhizodeposits are likely to depend on the time after the last labelling pulse had been applied, with more $^{13}C$-enriched rhizodeposits being produced shortly

after labelling (Kuzyakov, 2006; Kuzyakov and Gavrichkova, 2010). Second, the fact that most carbon is allocated to belowground plant parts during the first 1 to 2 months after growth initiation (Pausch and Kuzyakov, 2018) likely causes an important temporal variation in the $^{13}C$ enrichment of roots. Recently, Stevenel et al. (2019) have shown that the $\delta^{13}C$ value of roots of isotopically labeled *Canavalia brasiliensis* differed both temporally and spatially, as different parts of the root had a different isotopic value at the end of their 22-day experiment.

To assess the uncertainty of calculated values of subsoil carbon sequestration, we assessed how these values differ when the value of root $\delta^{13}C$ is varied with +/- 25 % (Figure S5). This results in calculated values of total carbon rhizodeposition, down to a depth of 0.75 m, of 68 – 101 g m$^{-2}$ for Mont-Calme 268, 103 – 154 g m$^{-2}$ for Probus, 76 – 106 g m$^{-2}$ for Zinal and 87 – 124 g m$^{-2}$ for CH Claro, or uncertainties in the amount of carbon rhizodeposits between – 14 and + 28 %. Further research on the effect of the assumption of using root $\delta^{13}C$ values as a proxy for carbon rhizodeposits is thus necessary to better quantify the

effect on estimates of carbon sequestration.

**4.4 The effect of old and recent wheat cultivars on soil organic carbon stabilization**

The aim of this study was to assess the effect of wheat cultivars from a century of wheat breeding in Switzerland on belowground carbon allocation and SOC stabilization. We found that the old wheat cultivars, with deeper active roots throughout the experiment and larger root biomass, allocated more assimilated carbon belowground (Figure 4C, Table 2).

However, we found no evidence that wheat cultivars with larger root biomass lead to additional SOC stabilization, as both the lowest (85 ± 27 g m$^{-2}$ by Mont-Calme 268) and highest (126 ± 57 g m$^{-2}$ by Probus) amounts of carbon stabilization in the soil were observed for the wheat cultivars with the highest root biomass (Table 2). Our hypothesis, which stated that wheat cultivars with larger root biomass and deeper roots would lead to larger amounts of carbon stabilization, could therefore not be confirmed in the short term.

The total amount of OC that will be stabilized in the soil by the studied wheat cultivars will depend on the fate of the root biomass in the long-term. The root biomass was higher for the old wheat cultivars, although these differences were mainly limited to the upper 0.45 m of the soil. Due to the destructive sampling of vegetation and soil at the end of the experiment, the fate of root biomass after harvest could not be assessed. Based on the results, one could therefore hypothesize that the higher root biomass of old wheat cultivars would lead to larger rates of carbon sequestration in the long-term. Similarly, Mathew et

al. (2017) suggested that growing grasses and maize plants would lead to larger SOC stocks because these plants have the highest total and root biomass, compared to growing crops with a lower biomass. However, higher carbon stocks in plant biomass do not necessarily lead to higher rates of SOC sequestration, as it has been suggested that easily decomposable organic compounds, such as root exudates, lead to higher rates or OC sequestration compared to more complex organic compounds,





such as root biomass (Castellano et al., 2015; Cotrufo et al., 2013). High-quality carbon inputs, such as root exudates, are more

easily incorporated into the microbial biomass, of which the necromass is an important precursor of stabilized SOC (Denef et al., 2010; Kallenbach et al., 2016; Kästner and Miltner, 2018; Kögel-Knabner, 2002). Therefore, the portion of root biomass that is eventually stabilized in soils through interactions with minerals (Hemingway et al., 2019; Kleber et al., 2015) or incorporated in soil aggregates (Six et al., 2000) will determine how much of this carbon is sequestered over the long-term. The cultivars of wheat that are grown around the world have gone through considerable changes throughout the past century,

with a continuing increase in the importance of wheat cultivars with smaller root biomass (Fossati and Brabant, 2003; Friedli et al., 2019; Waines and Ehdaie, 2007). This can have profound implications for OC stocks of soils under wheat cultivation, as rhizodeposition and root-derived carbon are the most important inputs of OC to the soil (Kong and Six, 2010). Our results show that over the course of one growing season, the amount of net rhizodeposition of the old cultivars (on average $105 \pm 63$ g m$^{-2}$) was similar to the recent cultivars (on average $106 \pm 55$ g m$^{-2}$). In contrast, we observed differences in the total amount

of belowground carbon allocation to the roots and rhizodeposition combined between old (on average $171 \pm 35$ g m$^{-2}$) and recent (on average $141 \pm 28$ g m$^{-2}$) cultivars. This suggests that in the long-term, more recently developed wheat cultivars might lead to less SOC sequestration compared to older cultivars due to the lower root biomass of the former. Testing the long-term effect of the gradual change in wheat cultivars on OC inputs to the soil would thus require experiments that run over multiple growing seasons, and allow the quantification of the amount of root carbon that is eventually sequestered in the soil.

Correct knowledge on the amount of OC that is transferred belowground by plants is necessary to reliably model SOC dynamics. However, this knowledge is currently limited and changes in belowground carbon allocation due to the cultivation of different cultivars is generally not considered in SOC models. Moreover, it has recently been shown that accounting for changes in belowground carbon allocation by relating this to changes in aboveground biomass does not improve model results (Taghizadeh-Toosi et al., 2016). Rather, it has been suggested that more reliable model results are obtained when crop-specific

amounts of belowground carbon allocation are used, independent of aboveground biomass production (Taghizadeh-Toosi et al., 2016). Since model results are very sensitive to the amount of carbon inputs (Keel et al., 2017), and cereal crops are grown on ca. 20 % of croplands globally (Leff et al., 2004), a correct assessment of a potential decrease in belowground carbon inputs by wheat plants over the past century through the cultivation of different cultivars will have important implications for the simulation of changes in SOC on the global scale.

Assessing the overall impact of the past evolution of wheat cultivars on SOC stocks also requires taking into account the amount of land needed to produce sufficient food. For example, if future research would show that more recent wheat cultivars lead to less SOC stabilization compared to older cultivars, this does not necessarily imply a net loss of SOC as a consequence of the historical shift to planting recently developed wheat cultivars. Indeed, if the aim is to increase overall SOC stocks, it might be more favorable to grow high-yield wheat cultivars that sequester less OC per unit area compared to a low-yielding

cultivar, if this results in a larger area of arable land that can be taken out of cultivation. This land can be put under native vegetation, such as forest or grassland, which is known to store substantially more SOC compared to arable land (Jobbágy and

Jackson, 2000). In addition, high-yield recent wheat cultivars often have a higher N use efficiency, which might have positive effects on reducing N losses (Aziz et al., 2017; Brancourt-Hulmel et al., 2003; Voss-Fels et al., 2019).

## 5. Conclusion

In this study, four different wheat cultivars were grown in lysimeters and labeled with $^{13}CO_2$ using repeated pulse-labelling to quantify the effect of rooting depth and root biomass on carbon rhizodeposition and stabilization. Our results show that there is no clear trend between the time of cultivar development and the amount of carbon inputs into the soil that is stabilized. Our results confirmed Friedli et al. (2019), who showed that older wheat cultivars from the Swiss breeding program had a higher root biomass and rooted deeper compared to more recently released wheat cultivars. However, all wheat cultivars resulted in

similar amounts of net carbon rhizodeposition, with large variabilities being observed between replicates of the same cultivars. Based on these results, the hypothesis that wheat cultivars with a larger root biomass and deeper roots would promote carbon stabilization, was rejected. An important remaining uncertainty is related to the fate of root biomass after harvest, which might contribute to the stabilized SOC pool over the long-term.

### Data availability

Additional figures and tables can be found in the supplementary information. The data associated with this manuscript is available in the online version of this manuscript.

### Author contribution

CDK, SA, AH, CF and JS conceived the idea for the study

CDK set up the lysimeter and labelling experiments and collected the data

SG, CDK, SA and RW performed lab analyses

MVdB, SG, CDK and JS analysed and interpreted the data and performed the statistics

MVdB and SG wrote the manuscript, with contributions from CDK, AH, SA, JS, CF and RW

MVdB and SG contributed equally to this manuscript

### Conflict of interest

The authors declare that they have no conflict of interest.



## Acknowledgements

The lysimeter infrastructure for this study was funded by ETH core start-up funds provided to Johan Six. Charlotte Decock is grateful for funding from the Plant Fellows, a postdoctoral fellowship administered by the Zurich Basel Plant Science center and funded under the European Union's Seventh Framework Programme for research, technological development and demonstration under grant agreement no GA-2010-267243 – PLANT FELLOWS. Additional funding for this project came from the Swiss National Science Foundation (Project number 200021_160232). The authors are very grateful to Matti Barthel, Ben Wild and Chris Mikita, for their help with setting up the lysimeters, data collection and interpretation. The authors appreciate the grassland sciences group of ETH Zürich for providing the laboratory facility to perform part of the analyses performed in this study. We thank Brigitta Herzog and Hansueli Zellweger for their help with greenhouse management and plant protection at the ETH Research Station for Plant Sciences in Lindau.

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

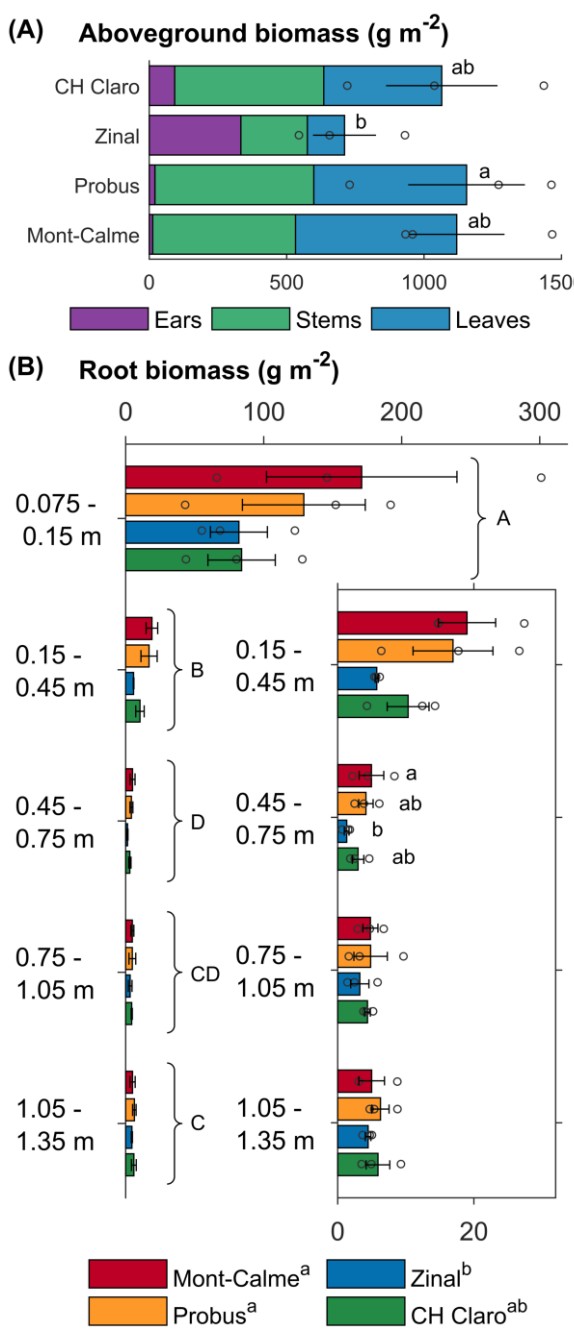


**Figure 1 Aboveground (A) and root (B) biomass of the different wheat cultivars at the end of the experiment. Bars represent the average per wheat cultivar, error bars show the standard error (n = 3) and circles show the individual data points. The inset in (B) shows a detail of the subsoil root biomass. If statistically significant differences were present, these are indicated with letters, with**
**variables sharing a letter not being significantly different. For root biomass, statistical analyses were performed using a three-way analysis of variance, in contrast to the two-way analysis used to calculated significant differences for the total root biomass summed over all depths as reported in Table 1 (see section 2.4).**





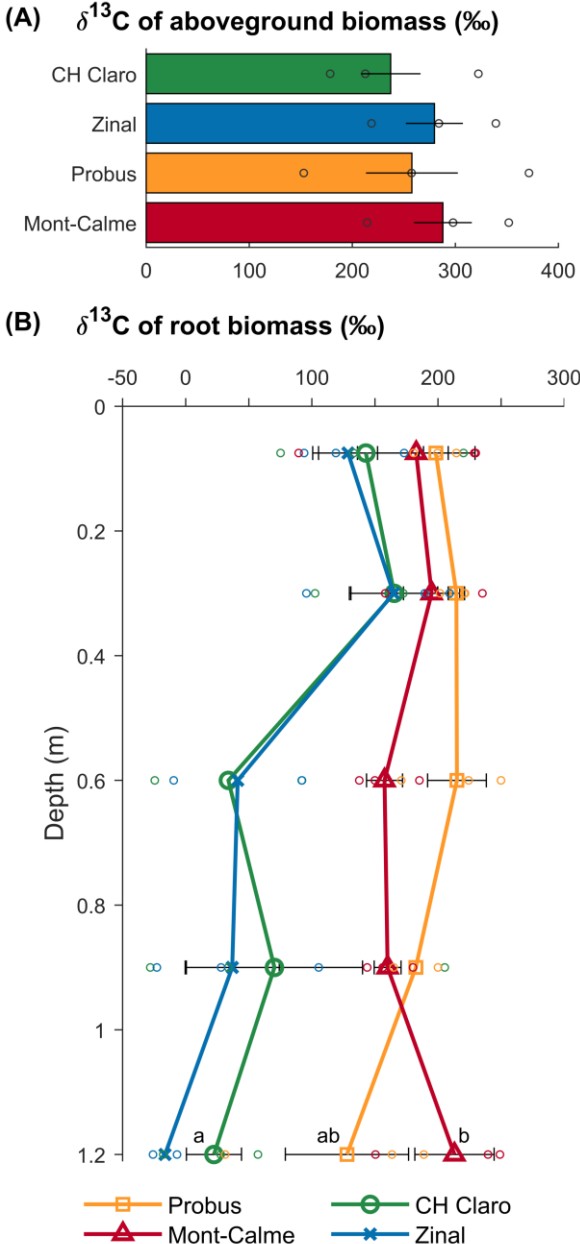

**Figure 2** $\delta^{13}$C value of aboveground (A) and belowground (B) biomass for the different wheat cultivars at the end of the experiment. Bars (A) and symbols (B) represent the average per wheat cultivar based on 3 replicates, error bars show the standard error (n = 3), while symbols without error bard indicate samples for which no 3 replicates were available. Circles show the individual data points. If statistically significant differences were present for root biomass at the same depth, these are indicated with letters, with variables sharing a letter not being significantly different and data points without error bars being left out of the analyses.



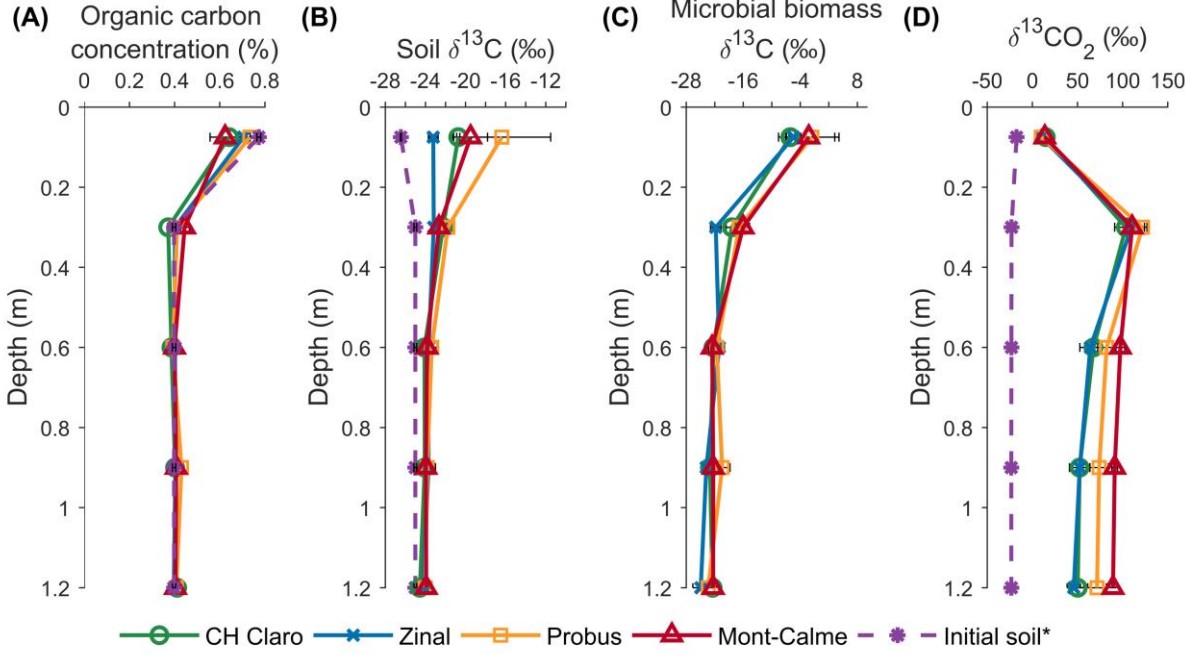


**Figure 3 Depth profiles of organic carbon concentration (A), the δ¹³C value of organic carbon (B), the δ¹³C value of microbial biomass (C) and the δ¹³C value of soil CO₂, averaged per wheat cultivar at the end of the experiment. Error bars represent the standard error (n = 3). *The initial soil indicates measurements performed on the soil that was used to fill the lysimeters prior to the experiments.**






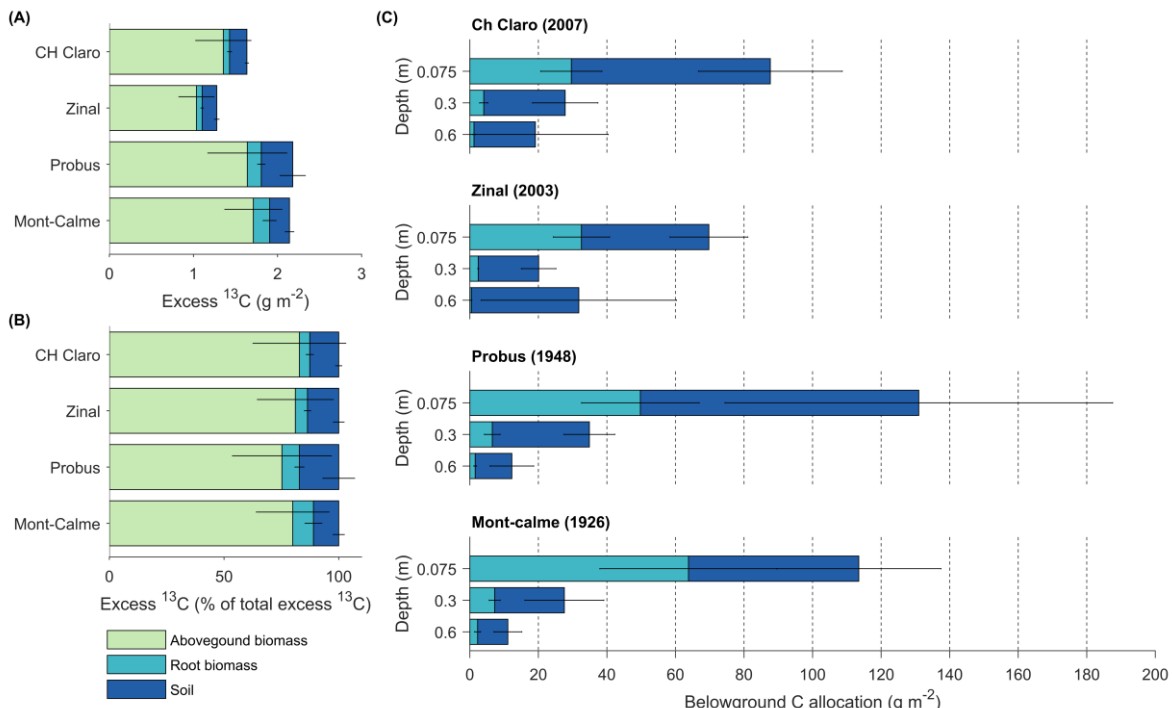

**Figure 4** Absolute (A) and relative (B) distribution of excess $^{13}$C between aboveground biomass, root biomass and soil for the different wheat cultivars. (C) shows depth profiles of total rhizodeposition carbon and root carbon for the upper three soil layers for the different wheat cultivars. Error bars represent the standard error (n = 3).


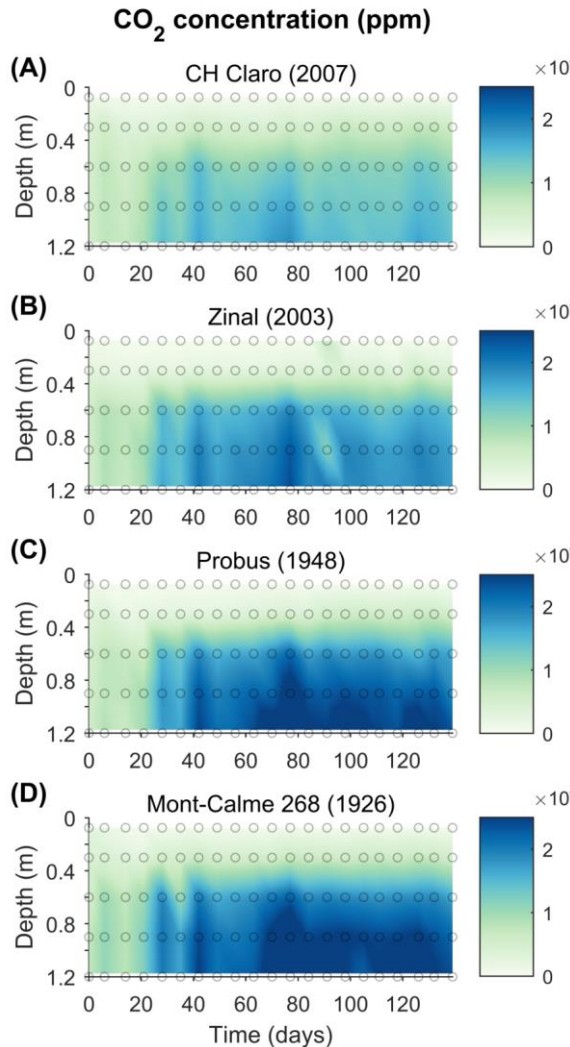

**Figure 5 Changes in the CO₂ concentration (ppm) in the lysimeters throughout the experiment for the four wheat cultivars. The average CO₂ concentration of three replicates are shown (n = 3). Dots indicate the measured data points.**





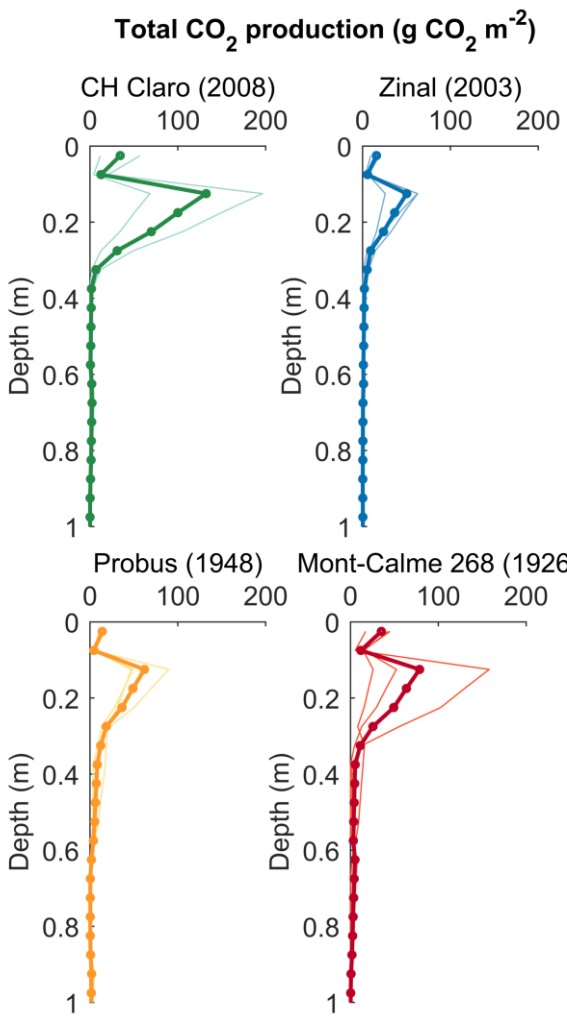

**Figure 6 Depth profiles of calculated cumulative CO₂ production (g CO₂ m⁻² per 0.05 m depth layer). Dots show the calculated production rates, thin lines show the calculated CO₂ production for the individual lysimeters, while the thick lines show the average based on three replicates (two for CH Claro).**





**Table 1 Characteristics (± standard error, n = 3) of the biomass of the different wheat cultivars at the end of the experiment. No significant differences were found between the belowground biomass (average biomass per cultivar summed over the different depths, using a two-way anova, see section 2.4) or root:shoot ratio of the different cultivars. Values that share a letter in the same column are not significantly different.**

| Cultivar (year of release) | Aboveground biomass | | Root biomass | | Root:shoot ratio |
| | Biomass (g m$^{-2}$) | OC % | Biomass (g m$^{-2}$) | OC % | |
|---|---|---|---|---|---|
| CH Claro (2007) | 1064 ± 207[ab] | 40.5 ± 0.3 | 107 ± 28[a] | 38.7 ± 1.9 | 0.10 ± 0.02[a] |
| Zinal (2003) | 710 ± 114[b] | 40.0 ± 0.14 | 97 ± 20[a] | 39.1 ± 1.1 | 0.14 ± 0.01[a] |
| Probus (1948) | 1154 ± 220[a] | 41.9 ± 0.2 | 161 ± 54[a] | 38.1 ± 0.5 | 0.13 ± 0.03[a] |
| Mont-Calme 268 (1926) | 1119 ± 174[ab] | 40.8 ± 0.4 | 205 ± 67[a] | 36.8 ± 1.3 | 0.19 ± 0.08[a] |


**Table 2 Average belowground carbon allocation (net rhizodeposition and root biomass combined) and net carbon rhizodeposition by the different wheat cultivars, calculated down to a depth of 0.75 m (variation is reported as the standard error, n = 3)**

| | Belowground C allocation (g m$^{-2}$) | Net carbon rhizodeposition (g m$^{-2}$) |
|---|---|---|
| CH Claro | 147 ± 40 | 112 ± 39 |
| Zinal | 135 ± 40 | 100 ± 39 |
| Probus | 184 ± 60 | 126 ± 57 |
| Mont-Calme 268 | 159 ± 37 | 85 ± 27 |
