# Peer review of "The soil organic carbon stabilization potential of old and new wheat cultivars: a 13CO2 labelling study"

_Biogeosciences, 2019_

## Referee Comment (RC1) · Stefan Karlowsky (Referee) · 31 Jan 2020

General Comments

In the present study, the authors report their findings from a 13CO2 pulse labelling experiment on different wheat cultivars grown in lysimeters filled with agricultural soil (surface and subsoil). The main study objective is to assess how the use of more recent wheat cultivars with lower rooting depths and root biomass alters organic carbon inputs into soil compared to older cultivars from the Swiss wheat breeding program. This research subject is important, because a large share of the global agricultural land is allotted to the cultivation of cereal crops, and it is unclear how the use of modern cultivars with altered root traits affect soil organic carbon (SOC) dynamics and consequently SOC stabilisation. Here the authors found no significant effects of different wheat cultivars on SOC in the short term. They conclude that the fate of root biomass after the harvest determines cultivar effects on stabilised SOC pools in the long term. The study is based on a sophisticated methodological approach, including an innovative lysimeter-labelling chamber setup as well as state-of-the-art 13C labelling and analysis techniques. The description of materials and methods used for the study is, in general, detailed enough to follow all steps of the experiment. However, a few things still need clarification (see specific comments). The major limitations of the study are the low number of replicates (probably due to the complex setup) and the fact that root biomass was too low for 13C analysis in many samples. Especially the latter impedes drawing conclusions about the input of plant-derived carbon into soil and its variation between the different cultivars over the growing season. Notably, the authors are well aware of these limitations and discuss them appropriately. The presentation of results is generally OK but should be modified in order to avoid redundancy between figures and tables. E.g. Fig. 1 and Table 1, both are showing the same values and statistics for aboveground biomass. Furthermore, you do not need to repeat values shown in figures and tables in the text body, neither in the results nor in the discussion section. I would also recommend to change Fig. 1 and Fig. 2, not separating between biomass and delta13C, instead showing aboveground biomass together with its delta 13C in Fig. 1 and the root parameters in Fig. 2 (as it is structured in the text body). Regarding Fig. 4 and Table 2, I am missing the statistics. These statistics would be necessary to support some of your interpretations from the discussion part. The discussion itself is a bit lengthy and would profit from some restructuring (see also specific comments). The subsections 4.1 to 4.3 can be shortened, e.g. by excluding the repetition of results and streamlining the remaining text. Maybe it is also better to start the discussion section with the main study object (for non-expert readers), which suddenly comes up in subsection 4.4 now. Another possibility to increase readability would be the use of more active and less passive voice, though this is a matter of taste. Overall, the

structure of the manuscript is clear and the language is fine. The authors relate their work to a comprehensive set of up-to-date literature and make the data underlying the results available as supplementary material. However, there are a few things in need of improvement and the manuscript will profit from a revision taking into account the addressed points.

Specific Comments

Line25: I think that "net SOC stabilization" is the wrong term. Stabilization implies a long-term effect, which you did not study here (if there was a difference - what about rhizodeposits degraded and respired by microbes off-season?). Therefore, better use "net carbon rhizodeposition" as in the rest of the manuscript.

Line 85: To my mind, this sentence is unnecessary, because the rationale of the study should be clear from the text above. I suggest starting directly with your research questions and marking them as such.

Line 143: Please indicate the approximate time of day when the labelling was carried out.

Line 146: Was it always the same chamber/cultivar for monitoring $CO_2$ concentrations?

Line 149: Is there an estimate for the $CO_2$ concentration at the end of the two hours?

Line 158: What does "limited amount of samples" mean – only at the end of the experiment (i.e. data shown in Fig. 3)?

Line 175: From my own experience, it is better to analyse soil microbial biomass directly from fresh (unfrozen) soil, because the freezing can increase the amount of carbon found in the non-fumigated fraction (probably cell lysis). However, regarding the delta13C values in comparison to SOC, this does not seem to be a problem here.

Line 211: Did you use the same value of -28 ‰ for aboveground biomass?

Lines 217-218: This sentence is unclear. With "some of the input variables", do you

mean biomass or delta13C?

Line 225: Please explain why you used the Janzen and Bruisma's equation in addition to excess 13C. If I understand it correctly, Fig. 4A shows the summed values for all soil layers as excess 13C according to Eq. 3 and Fig. 4C shows the data for individual soil layers as rhizodeposition C according to Eq. 4. However, the unit in Fig. 4C (g m-2) rather points to excess 13C. This must be clarified.

Line 284: Did you find a significant effect for the three blocks? Why did you use the blocks as fixed effects and not as random effects, i.e. error term, in the ANOVA? Please also report the significance levels for the different statistical tests. In general, I would prefer using the Tukey-HSD test, because it also accounts for multiple comparisons (in particular when depth is added as additional factor).

Lines 296-300: The aboveground biomass values are repeatedly reported in the text, Fig. 1A and Table 1/Table S1. It is sufficient to show the results once, especially since all individual values are available in the supplementary excel file. Remove this redundancy.

Line 325: Interpretations/conclusions do not belong to the results section. Delete this sentence.

Line 341: Note that the soil microbial biomass was higher in Zinal (Fig. S3), so that excess 13C was probably similar to Mont-Calme 268 (Fig. 4C).

Line 357: Do you mean "statistically significant" with "substantially"? Unfortunately, no statistical information is provided in Fig. 4.

Lines 364-367: Please improve the sentence structure.

Line 372: Do you have any explanation for the abrupt increase of Co2 concentrations?

Line 399: How are your results (no differences in root biomass between cultivars) in line the study of Friedli et al. (2019), showing substantially (statistically significant?)

higher root biomass in older cultivars than in more recent ones? That is contradictory!

Line 406: To which species does the root:shoot ratio of 0.14 belongs to, is it an average value?

Lines 429-444: This paragraph reads like an introduction passage. It is better to move it to delete it from the discussion and combine it with overlapping parts of the introduction.

Lines 459-471: The repetition of results should be avoided and the two paragraphs streamlined to 2-3 short sentences.

Line 492: By "assess the effect of wheat cultivars from a century of wheat breeding", do you mean that you assessed the effect of four cultivars representative for changes during a century of wheat breeding?

Line 494: There is no statistical support for this statement, neither for root biomass nor for belowground carbon allocation. In consequence, it is not surprising that you did not find effects on the SOC pool according to the next sentence.

Lines 506-509: This cannot be generalized, because the activity and substrate preference of microbial communities depends on a variety of factors (e.g. Delgado-Baquerizo et al., 2016: https://doi.org/10.1111/1462-2920.13642). In addition, the preference for recent plant-derived substrates or more stable SOM varies with soil depth (Kramer & Gleixner, 2008: https://doi.org/10.1016/j.soilbio.2007.09.016) and the presence/quality of plant residues is known to alter soil microbial communities (e.g. Bai et al, 2016: http://dx.doi.org/10.1016/j.apsoil.2015.09.009). In this sense, the microbial community can be shifted to more fungi and Gram-positive bacteria in the presence of more complex organic compounds derived from root residues.

Line 519: There is no statistically significant difference, only a slight trend.

Technical Corrections

Lines 47-49: Please reformulate the two passages with "is also proposed". This is very

repetitive, since the term "has been proposed" is already present in Lines 45-46.

Line 146: Obviously, this should be 40 g and not 40 mg.

Line 179: This sentence fits better in the previous subsection at line 172.

Line 193: Technically, you measured the delta 13C of C-F and C-NF instead of microbial biomass.

Lines 249-250: Separate the "s" and "i-1,i"/"i,i+1" in the formulas (maybe by a semicolon), as it can be confusing otherwise.

Line 350: Replace "showed substantial variation" with "varied".

Line 381: Include "biomass" (Plant biomass, carbon dynamics...).

Line 416 "were respiring CO2 down to greater depths..." -> Reformulate.

Line 453: The shown references do not include only the same studies.

Line 485: Twice "assess/ed"

---

## Referee Comment (RC2) · Anonymous Referee #2 · 18 Feb 2020

General comments

The authors address the question whether the aim of modern plant breeding strategies to maximize grain yield may affect soil C dynamics, because these strategies often have the side effect of reduced root biomass and reduced rooting depth. It is a relevant topic, because optimization of C storage in arable lands can contribute to higher soil C storage and better soil functions. The study has the potential to deduce recommendations for a climate-smart agricultural practice.

The authors conducted a 13C pulse-labelling mesocosm experiment with four wheat cultivars in a closed-chamber greenhouse setting. The elaborate experimental design

allows to quantify not only C input, but also soil respiration within a depth profile and is therefore highly suitable to address the question of subsoil C turnover. However, the amount of root biomass in deeper layers was in some cases too low to perform isotopic analyses, which then hampers drawing conclusions on subsoil C input. The thorough experimental set-up comes with the cost of low replication, which may be a reason for the high variability many of the measured parameters exhibited.

Throughout the paper the authors refer to C stabilization, but they actually did not quantify this (rather long-term) process. As in the title, they rather deduce the potential of SOC stabilization from other processes. Therefore, I suggest referring to the (rather short-term) processes that were actually investigated, which were root growth, rhizodeposition and soil C dynamics. Also, they do not present any results on C stabilization itself (which might be challenging given the duration of the experiment) and do not mention this concept in the introduction extensively. I suggest to adapt the terminology to achieve a more precise and coherent wording in a revised version of this manuscript.

Specific comments

Introduction:

The introduction gives a good overview, but could be more concise: Shorten and/or combine paragraphs 2 and 3.

Lines 61f: Please comment on processes that lead to differences in gross and net rhizodeposition. Are there studies on qualitative differences of rhizodepositions between wheat cultivars?

Lines 74f: A bit vague, which practical limitations do you mean?

Lines 90f: Hypothesis: Does the experimental design allow to test C stabilization or rather C input/ C balance? Soil C stabilization mechanisms are not assessed, only inferred from other processes (e.g. root growth, rhizodeposition). As far as I understood your study, you did not differentiate between different soil C forms, e.g. mineral

associated carbon or labile carbon that are a proxy for C stabilization.

Methods:

Well written, good level of detail, mostly easy to follow.

Lines 118f: Why did you choose a cultivar with known high rooting depth? Is this still characteristic for the group of "new" cultivars, or would this be a specific, maybe drought-adapted, cultivar? Since you argue with the two groups of "old" and "new" cultivars later on, I expect your selected cultivars not to be much different from commonly used cultivars.

Line 122: Was this the same topsoil as in the lysimeters?

Lines 127f: Did you measure plant biomass per seedling before transplanting/ labelling? What was the phenological stage of the seedlings? Did it differ between individuals and/ or cultivars?

Line 144: Do you mean $CO_2$ concentration of 58% or 58 atom% $^{13}CO_2$?

Line 158: Please be more specific, what does "limited amount" mean?

Line 176: 40mg, isn't this very low? Rather 40g, with 200 ml?

Line 176: Was the chloroform ethanol-free?

Lines 217f: Which input variables do you mean specifically?

Results:

Lines 295ff/ Section 3.1: Is stem and leaf biomass so much lower in Zinal because of much earlier grain filling? Please include data on phenological states for all cultivars or state more clearly if they have been in the same phenological stage (which I assume they have not). You only mention that they all reached flowering stage, but this does not exclude some been even further developed.

Lines 327ff. Do you expect SOC in the initial soil to differ from SOC in the soil in the

lysimeters at the begin of the experiment? If so, how?

Line 345: $CO_2$, not $d^{13}CO_2$ (A value cannot be enriched)

Lines 368ff: What about the $d^{13}C$ of $CO_2$ that was measured in some samples?

Figure 1: Using the same colors for different groups is confusing (e.g. leaves in 1A vs. Zinal in 1B). I don't find the inset in 1B useful, the statistics could be included in the main figure.

Figure 2, 3: Please use your color coding also for error bars.

Figure 3A: Use an x-axis range that fits the data, starting higher than 0.

Table 1 is redundant, except for root:shoot ratios.

Discussion:

Please do not repeat values, except for comparisons with other studies, where you name their values explicitly. Also, please do not only repeat results.

Line 399: Which differences in root architecture do you have in mind?

Lines 424ff: $d^{13}C$ in roots, variation with depth and cultivar: Why would the $d^{13}C$ signal of plant carbon change throughout the experiment, given that all plants received $^{13}CO_2$ in regular time periods and equal amounts. Do you expect seedling biomass at the time of transplanting/ before the first labelling to differ and therefore causing these differences? If you started with equal plant biomass and equal amounts of $^{13}C$, I would not expect these strong differences in plant $d^{13}C$.

Lines 431-444: This paragraph is appropriate as part of the introduction, rather than the discussion.

Lines 503ff: Could you test this hypothesis with your data for short-term relationships, independently from cultivar development time, e.g. by looking for relationships between SOC concentration and root biomass or $d^{13}C$ in soil and SOC concentrations?

[Figure]

Lines 531f: Please also mention the share of croplands in total landmass and the share of SOC of croplands in global SOC to give a comprehensive perspective.

Technical corrections

Line 786: Error bar (or bars), not bard

---

## Author Comment (AC1) · 19 Mar 2020

**Replies on referee comments on Van de Broek et al., The soil organic carbon stabilization potential of old and new wheat cultivars: a $^{13}$CO2 labelling study**

**Replies to Stefan Karlowsky**

We would like to thank Dr. Karlowsky for his comprehensive and detailed comments on our manuscript. These will greatly improve the quality of our manuscript. We present the reviewer comments in *italic*, our replies are formulated in normal font.

**General Comments**

*In the present study, the authors report their findings from a 13CO2 pulse labelling experiment on different wheat cultivars grown in lysimeters filled with agricultural soil (surface and subsoil). The main study objective is to assess how the use of more recent wheat cultivars with lower rooting depths and root biomass alters organic carbon inputs into soil compared to older cultivars from the Swiss wheat breeding program. This research subject is important, because a large share of the global agricultural land is allotted to the cultivation of cereal crops, and it is unclear how the use of mod ern cultivars with altered root traits affect soil organic carbon (SOC) dynamics and consequently SOC stabilisation. Here the authors found no significant effects of different wheat cultivars on SOC in the short term. They conclude that the fate of root biomass after the harvest determines cultivar effects on stabilised SOC pools in the long term. The study is based on a sophisticated methodological approach, including an innovative lysimeter-labelling chamber setup as well as state-of-the-art 13C labelling and analysis techniques. The description of materials and methods used for the study is, in general, detailed enough to follow all steps of the experiment. However, a few things still need clarification (see specific comments). The major limitations of the study are the low number of replicates (probably due to the complex setup) and the fact that root biomass was too low for 13C analysis in many samples. Especially the latter impedes drawing conclusions about the input of plant-derived carbon into soil and its variation between the different cultivars over the growing season. Notably, the authors are well aware of these limitations and discuss them appropriately.*

*The presentation of results is generally OK but should be modified in order to avoid redundancy between figures and tables. E.g. Fig. 1 and Table 1, both are showing the same values and statistics for aboveground biomass.*

The data were presented in the figure for visual interpretation, while we repeated the values in the table so the exact values are available to the reader. In order to reduce further redundancy, we no longer cite these values in the text, but refer to the table.

*Furthermore, you do not need to repeat values shown in figures and tables in the text body, neither in the results nor in the discussion section.*

These values are now removed from the text.

*I would also recommend to change Fig. 1 and Fig. 2, not separating between biomass and delta13C, instead showing aboveground biomass together with its delta 13C in Fig. 1 and the root parameters in Fig. 2 (as it is structured in the text body).*

We thank the reviewer for this suggestion, but we think it's more convenient for the reader to see separate figures for OC % and $\delta^{13}$C, although this is discussed in the text differently. This way, for example, we aim to emphasize the important differences in the $\delta^{13}$C value of above- and belowground biomass in the subsoil between the old and more recent cultivars.

*Regarding Fig. 4 and Table 2, I am missing the statistics. These statistics would be necessary to support some of your interpretations from the discussion part.*

We fully agree with the reviewer that statistics would aid our interpretation and discussion of the results. However, as explained at the end of section 2.2.4, due to (i) the large variability in root $\delta^{13}$C among the different replicates and (ii) the low biomass of retrieved roots, which prevented $\delta^{13}$C analyses for roots at certain depths for multiple lysimeters, we decided to calculate net rhizodeposition using the average values for all 3 replicates.

The variations on the final calculated values (as shown using error bars in Figure 4) was calculated by error propagation calculation, using the standard errors of values used for these calculations when they were available (i.e. when values were present for the three replicates of a certain cultivars, for example for SOC %, bulk density etc.). This is now explicitly stated in the caption of Figure 4: 'Error bars represent the standard error (n = 3), which was calculated as error propagation based on the standard errors for the average for the different cultivars. This prevented statistical analyses of significant differences between the cultivars.'.

Therefore, the variation within the cultivars was not be accounted for, which prevented us from performing statistical analyses here. We are aware that this limits our interpretations, but acknowledge this in the text (section 2.3.1), where the following sentence has now been added: 'Uncertainties on these calculations were assessed using error propagation of the variables for which standard errors could be calculated (i.e., for which values were available for the three replicates of a cultivar). When standard errors of the $\delta$13C value of root biomass for a certain depth layer could not be calculated due to a low number of replicates, the standard error was calculated by multiplying the average relative standard error of the layers above and below this layer with the $\delta$13C value of this layer.'.

*The discussion itself is a bit lengthy and would profit from some restructuring (see also specific comments).*

We used your comments below to shorten the discussion

*The subsections 4.1 to 4.3 can be shortened, e.g. by excluding the repetition of results and streamlining the remaining text. Maybe it is also better to start the discussion section with the main study object (for non-expert readers), which suddenly comes up in subsection 4.4 now.*

Thanks for this suggestion. We now start the discussion by briefly repeating the objective of our study and the main results, for readers who jump to the discussion section at once. Where possible, we shortened sections 4.1 to 4.3, e.g. by removing the first paragraph of section 4.2.

*Another possibility to increase readability would be the use of more active and less passive voice, though this is a matter of taste. Overall, the structure of the manuscript is clear and the language is fine. The authors relate their work to a comprehensive set of up-to-date literature and make the data underlying the results available as supplementary material. However, there are a few things in need of improvement and the manuscript will profit from a revision taking into account the addressed points.*

**Specific Comments**

*Line25: I think that "net SOC stabilization" is the wrong term. Stabilization implies a long-term effect, which you did not study here (if there was a difference - what about rhizodeposits degraded and respired by microbes off-season?). Therefore, better use "net carbon rhizodeposition" as in the rest of the manuscript.*

A similar comment was raised by the second reviewer as well, and we agree with both reviewers. Therefore, we changed the term 'carbon stabilization' to 'net rhizodeposition' throughout the text. We, however, did not change the title of the manuscript, as here we talk about 'carbon stabilization potential', and net rhizodeposition and root biomass (which we study) given an indication about the potential to stabilize carbon on the longer term.

*Line 85: To my mind, this sentence is unnecessary, because the rationale of the study should be clear from the text above. I suggest starting directly with your research questions and marking them as such.*

Thanks for this suggestion. We removed these sentences and explicitly formulated the research question and the hypothesis.

*Line 143: Please indicate the approximate time of day when the labelling was carried out.*

The labelling was carried out at 2 pm, this has now been added to the text.

*Line 146: Was it always the same chamber/cultivar for monitoring CO2 concentrations?*

The monitoring was always done at the same chamber and thus cultivar. The monitoring intended to approximate general CO2 uptake within for instance changing chamber volumes and less to adjust for each individual cultivar/chamber. We agree with the reviewer that this is not ideal but we had to consider technical implementations as well as time issues. Therefore it was decided to only monitor at one chamber. However, given the relatively similar enrichment across all cultivars in aboveground plant biomass we believe that the labeling was done relatively homogeneous. We added to the text that $CO_2$ concentrations were always measured in the same chamber: 'Throughout the experiment, CO2 concentrations were measured in the same chamber.'.

*Line 149: Is there an estimate for the CO2 concentration at the end of the two hours?*

No, the $CO_2$ concentration in the chambers was not measured after these two hours.

*Line 158: What does "limited amount of samples" mean – only at the end of the experiment (i.e. data shown in Fig. 3)?*

Yes, that is what we meant. We clarified this in the text: 'In addition, the $\delta^{13}C$ value of $CO_2$ was measured for $CO_2$ samples collected along the depth profiles on the last sampling date, using a Gasbench II […]'

*Line 175: From my own experience, it is better to analyse soil microbial biomass directly from fresh (unfrozen) soil, because the freezing can increase the amount of carbon found in the non-fumigated fraction (probably cell lysis). However, regarding the delta13C values in comparison to SOC, this does not seem to be a problem here.*

We are aware of the fact that this would indeed be a better practice. However, due to technical constraints we had to perform the measures on frozen soil samples.

*Line 211: Did you use the same value of -28 ‰ for aboveground biomass?*

We only had to make an assumption about the $\delta^{13}C$ value of roots, which was necessary to calculate the excess $^{13}C$, to eventually calculate net rhizodeposition. As this was not done for aboveground biomass, we did not need to make assumptions about the $\delta^{13}C$ value of the unlabelled aboveground biomass.

*Lines 217-218: This sentence is unclear. With "some of the input variables", do you mean biomass or delta13C?*

This was related to the $\delta^{13}C$ of root biomass. This has now been added to the text: 'In addition, there was a large variability in the $\delta^{13}C$ value of root biomass between the replicates of the same cultivar, which complicated the calculation of excess $^{13}C$ for individual lysimeters.'

*Line 225: Please explain why you used the Janzen and Bruisma's equation in addition to excess 13C. If I understand it correctly, Fig. 4A shows the summed values for all soil layers as excess 13C according to Eq. 3 and Fig. 4C shows the data for individual soil layers as rhizodeposition C according to Eq. 4. However, the unit in Fig. 4C (g m-2) rather points to excess 13C. This must be clarified.*

That is correct: Eq. 3 was used to calculate the mass of recovered $^{13}C$ label (g $^{13}C$ m$^{-2}$), while Eq. 4 was used to calculate the total amount of net carbon rhizodeposition, using the excess $^{13}C$ in roots and the soil (g C m$^{-2}$). To make this more clear in Figure 4, the unit of the label of Fig 4a has been changed to (g $^{13}C$ m$^{-2}$), while the unit in the label if Fig 4c has been changed to (g C m$^{-2}$).

*Line 284: Did you find a significant effect for the three blocks? Why did you use the blocks as fixed effects and not as random effects, i.e. error term, in the ANOVA? Please also report the significance levels for the different statistical tests. In general, I would prefer using the Tukey-HSD test, because it also accounts for multiple comparisons (in particular when depth is added as additional factor).*

For some of the variables, we did find a significant effect of the blocks (e.g. aboveground biomass), while for other variables this was not the case (e.g. belowground biomass). For the analysis of statistical differences between properties of the cultivars (e.g. aboveground biomass), we used a two-way anova without interactions. This is generally recommended for the analysis of randomized complete block designs (e.g. Dean et al. (eds.), Handbook of design and analysis of experiments, ISBN 978-1-4665-0434-9, or https://stat.ethz.ch/~meier/teaching/anova/block-designs.html). Therefore, block was not treated as a random effect. We note that for the three-way anova, we included block as a random effect (see L. 289 – 291). The significance level for the Tukey's test is now added to this section: '[...] using a significance level of 0.05'.

*Lines 296-300: The aboveground biomass values are repeatedly reported in the text, Fig. 1A and Table 1/Table S1. It is sufficient to show the results once, especially since all individual values are available in the supplementary excel file. Remove this redundancy.*

Thanks for this suggestion. As stated above, we removed the values for aboveground biomass throughout the test. However, we prefer to show the values for aboveground biomass in Table 1, to give the reader a complete overview of the values of both above- and belowground biomass. We are aware of the fact that these values are shown in Figure 1, but we want the reader to be able to consult the exact values without having to go look for the online supplement.

*Line 325: Interpretations/conclusions do not belong to the results section. Delete this sentence.*

This sentence has been deleted.

*Line 341: Note that the soil microbial biomass was higher in Zinal (Fig. S3), so that excess 13C was probably similar to Mont-Calme 268 (Fig. 4C).*

Thanks a lot for this remark, we now included after that sentence: 'However, as the microbial biomass under Zinal was substantially higher compared to under Mont-Calme 268 in this layer, this does not necessarily imply that microbes under Mont-Calme 268 incorporated more excess $^{13}$C compared to under Zinal'.

*Line 357: Do you mean "statistically significant" with "substantially"? Unfortunately, no statistical information is provided in Fig. 4.*

We meant substantially, since no statistical test could be performed (see above)

*Lines 364-367: Please improve the sentence structure.*

We changed these sentences to: 'The total amount of net carbon rhizodeposition measured at the end of the experiment down to 0.75 m decreased with depth for all wheat cultivars, with the exception of Zinal (Figure 4C). The highest values were observed for Probus (126 ± 57 g C m$^{-2}$), followed by CH Claro (112 ± 39 g C m$^{-2}$), Zinal (100 ± 39 g C m$^{-2}$) and Mont-Calme (85 ± 27 g C m$^{-2}$). There was thus no clear relationship between the amount of net carbon rhizodeposition and year of release of the wheat cultivars.'

*Line 372: Do you have any explanation for the abrupt increase of Co2 concentrations?*

We think this was caused by roots growing down to these depths at this moment, although we do not have conclusive evidence for this. For this reason, we do not elaborate on this in the manuscript.

*Line 399: How are your results (no differences in root biomass between cultivars) in line the study of Friedli et al. (2019), showing substantially (statistically significant?) higher root biomass in older cultivars than in more recent ones? That is contradictory!*

As stated in line 395 – 396, we did find differences in root biomass between old (161 & 205 g m$^{-2}$) and recent (107 & 97 g m$^{-2}$) wheat cultivars, although these were not statistically significantly different (due to large variations within cultivars). Friedli et al. (2019) found that cultivars from the Swiss wheat breeding program showed decreasing root biomass with increasing year of cultivar development. Therefore, we state that our results are 'in line' with the results from Friedli et al. However, we have emphasized that are results are not statistically different (L 397 - 399).

*Line 406: To which species does the root:shoot ratio of 0.14 belongs to, is it an average value?*

This indeed is the average value for all the cultivars studied by Friedli et al.. This has been clarified in the text: '[...] including the cultivars used in our study (an average value of 0.14 for all cultivars studied by Friedli et al. (2019)).'

*Lines 429-444: This paragraph reads like an introduction passage. It is better to move it to delete it from the discussion and combine it with overlapping parts of the introduction.*

This comment was also raised by the other reviewer and we agree that this paragraph is redundant here. Therefore, we deleted this paragraph to reduce the length of the discussion, as part of this is covered in the introduction.

*Lines 459-471: The repetition of results should be avoided and the two paragraphs streamlined to 2-3 short sentences.*

Thanks for this suggestion, these 2 paragraphs can indeed be shortened considerably. We chose to retain the values we provide about the total amount of carbon that is allocated belowground, as this is not reported elsewhere in the manuscript, so we can compare them to literature values.

*Line 492: By "assess the effect of wheat cultivars from a century of wheat breeding", do you mean that you assessed the effect of four cultivars representative for changes during a century of wheat breeding?*

Yes, that is indeed what we meant. As we now re-stated the aim of our study at the beginning of the discussions (see above), we removed this sentence here, as it is redundant.

*Line 494: There is no statistical support for this statement, neither for root biomass nor for belowground carbon allocation. In consequence, it is not surprising that you did not find effects on the SOC pool according to the next sentence.*

See about the inability to statistically prove this in previous responses. We did find that there were no statistically significant differences between the root biomass of different cultivars (Table 1), although the averages suggest that root biomass was larger for the older cultivars. To clarify this, we included '[...] allocated more assimilated carbon belowground, although this could not be statistically proven'.

*Lines 506-509: This cannot be generalized, because the activity and substrate preference of microbial communities depends on a variety of factors (e.g. Delgado-Baquerizo et al., 2016:*

*https://doi.org/10.1111/1462-2920.13642). In addition, the preference for recent plant-derived substrates or more stable SOM varies with soil depth (Kramer & Gleixner, 2008: https://doi.org/10.1016/j.soilbio.2007.09.016) and the presence/quality of plant residues is known to alter soil microbial communities (e.g. Bai et al, 2016: http://dx.doi.org/10.1016/j.apsoil.2015.09.009). In this sense, the microbial community can be shifted to more fungi and Gram-positive bacteria in the presence of more complex organic compounds derived from root residues.*

We agree with the reviewer that the fate of roots in the subsoil (mineralisation versus stabilization) is more complex than as we stated in the manuscript. Therefore, we shortened this section, as this is not the focus of our study, while briefly also incorporating the remarks raised by the reviewer: 'However, it is not straightforward to make predictions about the amount of root biomass that will be stabilized in the soil in the long term, as this depends on the efficiency with which plant-derived biomass is incorporated in microbial biomass (Cotrufo et al., 2013) and interactions between soil depth, the microbial community composition and its substrate preference (e.g. Kramer and Gleixner, 2008), among other factors.'.

*Line 519: There is no statistically significant difference, only a slight trend.*

We changed this sentence to: 'In contrast, despite the lack of statistical evidence, we observed differences […]'.

**Technical Corrections**

*Lines 47-49: Please reformulate the two passages with "is also proposed". This is very repetitive, since the term "has been proposed" is already present in Lines 45-46.*

Thanks for noticing this, we removed 2 of the 3 'is proposed' by an alternative wording.

*Line 146: Obviously, this should be 40 g and not 40 mg.*

I assume you meant line 176? Here, is should indeed be 40 g, thanks for noticing this.

*Line 179: This sentence fits better in the previous subsection at line 172.*

This sentence is at this location because the determination of the gravimetric moisture content was necessary to calculate microbial biomass carbon per unit dry soil. To make this clear to the reader, this sentence was changed to: 'To determine microbial biomass carbon per unit of dry soil, the gravimetric soil water content was determined by drying about 10 g of each soil sample at 105 °C and subtracting the weights before and after drying.'. We note that we also determined the gravimetric soil moisture content for bulk soil samples collected from the lysimeters at the end of the experiment. This is mentioned in line 171.

*Line 193: Technically, you measured the delta 13C of C-F and C-NF instead of microbial biomass.*

Thanks for pointing this out, we changed this in the manuscript: '[…] fumigated and non-fumigated soil (for the determination of microbial biomass C and $\delta^{13}C$) […]'.

*Lines 249-250: Separate the "s" and "i-1,i"/"i,i+1" in the formulas (maybe by a semicolon), as it can be confusing otherwise.*

Thanks for this suggestion, we changed this accordingly.

*Line 350: Replace "showed substantial variation" with "varied".*

This has been changed

*Line 381: Include "biomass" (Plant biomass, carbon dynamics…).*

This has been changed

*Line 416 "were respiring CO2 down to greater depths…" -> Reformulate.*

This was reformulated to '[…] roots of the old wheat cultivars respired $CO_2$ at greater depths compared to […]'.

*Line 453: The shown references do not include only the same studies.*

Thanks for pointing this out. We now shortened and combined both sentences, without mentioning the 'same studies'.

*Line 485: Twice "assess/ed"*

Thanks, this has been replaced

---

## Author Comment (AC2)

**Replies on referee comments on Van de Broek et al., The soil organic carbon stabilization potential of old and new wheat cultivars: a $^{13}$CO2 labelling study**

**Replies to reviewer 2**

We would like to thank the second reviewer for the comprehensive and detailed comments on our manuscript. These will greatly improve the quality of our manuscript. We present the reviewer comments in *italic*, our replies are formulated in normal font.

**General comments**

*The authors address the question whether the aim of modern plant breeding strategies to maximize grain yield may affect soil C dynamics, because these strategies often have the side effect of reduced root biomass and reduced rooting depth. It is a relevant topic, because optimization of C storage in arable lands can contribute to higher soil C storage and better soil functions. The study has the potential to deduce recommendations for a climate-smart agricultural practice.*

*The authors conducted a 13C pulse-labelling mesocosm experiment with four wheat cultivars in a closed-chamber greenhouse setting. The elaborate experimental design allows to quantify not only C input, but also soil respiration within a depth profile and is therefore highly suitable to address the question of subsoil C turnover. However, the amount of root biomass in deeper layers was in some cases too low to perform isotopic analyses, which then hampers drawing conclusions on subsoil C input. The thorough experimental set-up comes with the cost of low replication, which may be a reason for the high variability many of the measured parameters exhibited.*

*Throughout the paper the authors refer to C stabilization, but they actually did not quantify this (rather long-term) process. As in the title, they rather deduce the potential of SOC stabilization from other processes.*

*Therefore, I suggest referring to the (rather short-term) processes that were actually investigated, which were root growth, rhizodeposition and soil C dynamics. Also, they do not present any results on C stabilization itself (which might be challenging given the duration of the experiment) and do not mention this concept in the introduction extensively. I suggest to adapt the terminology to achieve a more precise and coherent wording in a revised version of this manuscript.*

Thanks for this comment. A similar concern was raised by the other reviewer. Therefore, we changed the term 'carbon stabilization' to 'net carbon rhizodeposition' throughout the text.

**Specific comments**

*Introduction:*

*The introduction gives a good overview, but could be more concise: Shorten and/or combine paragraphs 2 and 3.*

Thanks for this suggestion, we reduced the length of both paragraphs

*Lines 61f: Please comment on processes that lead to differences in gross and net rhizodeposition. Are there studies on qualitative differences of rhizodepositions between wheat cultivars?*

To clarify the difference between gross and net rhizodeposition, we now complemented this sentence with '[…] , after a portion of gross rhizodeposits are lost from the soil through microbial mineralization or leaching'. The only study we are aware of that has checked for differences in carbon rhizodeposition between different wheat cultivars is Hirte et al. (2018, https://doi.org/10.1016/j.agee.2018.07.010), to which we compare our results in the discussion section.

*Lines 74f: A bit vague, which practical limitations do you mean?*

Here, we refer to difficulties of continuously labelling agricultural crops in the field. We are aware of the existence of techniques that allow to continuously label the aboveground parts of plants in the field throughout the year (several FACE experiments do so), but this requires a very elaborate set-up with large financial investments. This is, however, not feasible for most studies. To make this clear, we changed this sentence to 'However, the continuous application of $13CO_2$ or $14CO_2$ during the course of an entire growing season to plants is often not feasible, as this requires the set-up of open-top chambers while continuously supplying the crops with the isotopic label, which comes at a high financial cost'.

*Lines 90f: Hypothesis: Does the experimental design allow to test C stabilization or rather C input/ C balance? Soil C stabilization mechanisms are not assessed, only inferred from other processes (e.g. root growth, rhizodeposition). As far as I understood your study, you did not differentiate between different soil C forms, e.g. mineral associated carbon or labile carbon that are a proxy for C stabilization.*

Our experimental set-up did indeed not allow us to check for long-term carbon stabilization through e.g. organo-mineral interactions. Therefore, we adapted our research question to 'do wheat cultivars with shallow roots and lower root biomass lead to less net carbon rhizodeposition compared to wheat cultivars with deeper roots and higher root biomass?' and our hypothesis to '[…] wheat cultivars with shallow roots and lower root biomass would result in less net rhizodeposition over the course of a growing season, compared to cultivars with deeper roots and higher root biomass.' In addition, throughout the manuscript we changed the term 'carbon stabilization' to 'net rhizodeposition', to make clear to the reader that we do not focus on long-term carbon stabilization, but rather on short term (one growing season) carbon rhizodeposition.

*Methods:*

*Well written, good level of detail, mostly easy to follow.*

*Lines 118f: Why did you choose a cultivar with known high rooting depth? Is this still characteristic for the group of "new" cultivars, or would this be a specific, maybe drought-adapted, cultivar? Since you argue with the two groups of "old" and "new" cultivars later on, I expect your selected cultivars not to be much different from commonly used cultivars.*

As our aim was to assess how wheat cultivars with different rooting depth affect belowground carbon dynamics, we specifically chose wheat cultivars with both low and high biomass. As stated in the manuscript (L. 117 - 118), rooting depth has generally increased through time in the Swiss wheat breeding program (Friedli et al. 2018; https://doi.org/10.1007/s10681-019-2404-7)) as well as in other wheat breeding programs (Aziz et al. 2017; DOI 10.1007/s11104-016-3059-y). For the Swiss wheat breeding program, Friedli et al. have shown that there is a consistent trend of decreased rooting depth with the year of cultivar development under well-watered conditions. Therefore, the

selected wheat cultivars are typical of the 'old' and 'more recent' cultivars developed by this breeding program.

*Line 122: Was this the same topsoil as in the lysimeters?*

This was indeed the same topsoil. We made this clear in the manuscript: 'Next, the seedlings were planted in containers filled with the same topsoil used to fill the lysimeters and transferred to […]'.

*Lines 127f: Did you measure plant biomass per seedling before transplanting/ labelling? What was the phenological stage of the seedlings? Did it differ between individuals and/ or cultivars?*

Plant biomass per seedling was measured before transplanting, but we observed no big differences between the cultivars. At the moment of transplanting, the plants were at the onset of tillering. The latter has been added to the manuscript (section 2.1.2): 'At the timing of transplanting, the plants were at the onset of tillering.'.

*Line 144: Do you mean CO2 concentration of 58% or 58 atom% 13CO2?*

We meant atom%, this has been changed in the manuscript.

*Line 158: Please be more specific, what does "limited amount" mean?*

We meant that we only performed $\delta^{13}C$ analyses of $CO_2$ for samples collected on the last sampling day. This has been adjusted in the text to make this clear to the reader: 'In addition, the δ13C value of CO2 was measured for CO2 samples collected along the depth profiles on the last sampling date […]'.

*Line 176: 40mg, isn't this very low? Rather 40g, with 200 ml?*

Thanks for noticing, this should indeed be 40 g, we changed this in the manuscript

*Line 176: Was the chloroform ethanol-free?*

Unfortunately, due to some limitations, the chloroform was not ethanol-free. However, since all samples were fumigated with the same chloroform, we think the comparison between different treatments and different depths is valid, as we mostly use these data in a qualitative way (e.g. comparison of microbes under which cultivar incorporated most $^{13}CO_2$)

*Lines 217f: Which input variables do you mean specifically?*

We meant the $\delta^{13}C$ value of root biomass. To make this clear to the reader, we changed this to: 'In addition, there was a large variability in the $\delta^{13}C$ value root biomass between the replicates of the same cultivar, which complicated the calculation of excess $^{13}C$ for individual lysimeters.'.

*Results:*

*Lines 295ff/ Section 3.1: Is stem and leaf biomass so much lower in Zinal because of much earlier grain filling? Please include data on phenological states for all cultivars or state more clearly if they have been in the same phenological stage (which I assume they have not). You only mention that they all reached flowering stage, but this does not exclude some been even further developed.*

We are not sure why stem and leaf biomass in Zinal was lower compared to the other cultivars. One reason might be that the Zinal cultivars were negatively affected by the growing conditions in the

greenhouse, although this was not the case for the other cultivars. We do not have detailed information about the phenological stage in which the plants were at the moment of harvest, so this can unfortunately not be included in the manuscript. We hope that the sentence in L. 300 – 301 ('It is noted that these data should be interpreted with care, since not all plants reached maturity at the time of harvest, and is potentially not representative for the biomass of the ears of full-grown plants.') will make clear to the reader that there are uncertainties with respect to the aboveground biomass.

*Lines 327ff. Do you expect SOC in the initial soil to differ from SOC in the soil in the lysimeters at the begin of the experiment? If so, how?*

Given the relative short duration of the experiment (ca. 5 months), we did not expect that the SOC concentration was different compared to the initial soil. Since we did not have a control treatment, we explicitly wanted to emphasize this lack of a difference in SOC concentration, which suggests that the SOC concentration was not affected by the type of cultivar.

*Line 345: CO2, not d13CO2 (A value cannot be enriched)*

Thanks for noticing this, we corrected this to: 'the $CO_2$ under the old wheat cultivars was more enriched in 13C compared to […]'

*Lines 368ff: What about the d13C of CO2 that was measured in some samples?*

In this section (3.5), we only present the calculated depth profiles of $CO_2$ in the different lysimeters. As we had only limited information about the $\delta^{13}CO_2$ depth profiles, while we did not had sufficient data to separate autotrophic from heterotrophic $CO_2$ production, we were not able to calculate the $\delta^{13}C$ value of produced $CO_2$ along the soil profile. Therefore, the $\delta^{13}C$ of $CO_2$ is not discussed in this section, as this has been done in section 3.3 (L. 343 – 347).

*Figure 1: Using the same colors for different groups is confusing (e.g. leaves in 1A vs. Zinal in 1B). I don't find the inset in 1B useful, the statistics could be included in the main figure.*

Thanks for pointing this out, we changed the colors in Fig. 1A. However, we prefer to keep the inset, as otherwise the values, and (lack of) differences between the root biomass at different depths would not be clear to the reader. That is also the reason why we put the letters indicating significant differences in the inset, so the reader can visually inspect the actual values behind the statistics, which would not be possible in the main figure.

*Figure 2, 3: Please use your color coding also for error bars.*

We would prefer to keep the error bars in black, because colored error bars would be much more difficult to differentiate from the data points, and will overlap with some of the individual data points that are plotted, which would make their colors less distinguishable.

*Figure 3A: Use an x-axis range that fits the data, starting higher than 0.*

To not give the impression that the organic carbon concentrations are very low, we prefer to keep starting the x-axis at 0.

*Table 1 is redundant, except for root:shoot ratios.*

Also the OC % of aboveground biomass, the total biomass and OC% of roots is not presented in the figures, so we would like to keep these numbers here so they can be consulted by the reader. To decrease redundancy, we no longer repeat these numbers in the text.

*Discussion:*

*Please do not repeat values, except for comparisons with other studies, where you name their values explicitly. Also, please do not only repeat results.*

Thanks for pointing this out, we no longer repeat these numbers in the text, except when we compare them to other studies, and we removed sections where we repeat the results while this was not necessary.

*Line 399: Which differences in root architecture do you have in mind?*

We meant mostly differences in root biomass. To clarify this to the reader, we changed this to: '(ii) actual differences in root biomass'.

*Lines 424ff: d13C in roots, variation with depth and cultivar: Why would the $\delta^{13}C$ signal of plant carbon change throughout the experiment, given that all plants received 13CO2 in regular time periods and equal amounts. Do you expect seedling biomass at the time of transplanting/ before the first labelling to differ and therefore causing these differences? If you started with equal plant biomass and equal amounts of 13C, I would not expect these strong differences in plant d13C.*

Our results indeed suggest that differences in timing of C allocation to deep roots (only shortly after planting for more recent cultivars versus throughout the growing season for old cultivars) cause the difference in the $\delta^{13}C$ value of roots. However, this does not need to imply that the $\delta^{13}C$ of plant biomass changed throughout the experiment. We interpreted this as follows: as more recent cultivars grew deep roots in the beginning of the experiment, less total $^{13}C$ had been assimilated by the plants over the period when root biomass was constructed. The old cultivars, which created root biomass throughout the experiment, therefore assimilated much more $^{13}C$ during the period when roots were build, therefore leading to higher $\delta^{13}C$ values of roots along the entire soil profile.

To make this clear to the reader, we changed part of this section to: 'This suggests that both old and more recent wheat cultivars grew roots down to depths of > 1 m in the beginning of the experiment (when the total amount of $^{13}C$ assimilated by the plants was limited), while only the old cultivars kept on allocating carbon down to deep roots (> 0.45 m) throughout the experiment (thus having assimilated more $^{13}C$ over the period of root growth compared to the more recent cultivars).'.

*Lines 431-444: This paragraph is appropriate as part of the introduction, rather than the discussion.*

Thanks for pointing this out, we now removed this paragraph.

*Lines 503ff: Could you test this hypothesis with your data for short-term relationships, independently from cultivar development time, e.g. by looking for relationships between SOC concentration and root biomass or d13C in soil and SOC concentrations?*

Testing this hypothesis would be difficult, given the relative short duration of our experiment. We suggest that the fate of decayed root biomass (thus upon incorporation in microbial biomass) will determine how much of this C will remain in the soil, mainly as stabilized microbial necromass. As we found no differences in SOC concentration between the cultivars, we do not expect to find relationships between root biomass and SOC concentration. Testing the hypothesis that higher root

biomass would lead to larger amounts of stabilized SOC was, however, outside the scope of our study.

*Lines 531f: Please also mention the share of croplands in total landmass and the share of SOC of croplands in global SOC to give a comprehensive perspective.*

Thanks for this suggestion, this has now been added to this section: '[…] and cereal crops are grown on ca. 20 % of croplands globally (Leff et al., 2004) (covering ca. 12 % of global land mass and storing ca. 10 % of global SOC in the upper meter of soil (Govers et al., 2013))  […]'.

***Technical corrections***

*Line 786: Error bar (or bars), not bard*

Thanks for noticing, this has been corrected

---

## Author Response (AR2)

Dear editor,

We would like to thank you and the two reviewers for having another look at our manuscript, and for accepting it. The feedback we received from both reviewers and you greatly improved the quality of this manuscript, and we are very grateful for this.

We followed the recommendation of Stefan Karlowsky to check for significant differences between excess $^{13}$C in the different compartments in Fig. 4(a). We found that these quantities did not differ significantly between any of the cultivars, and included this information in the caption of this figure.

During these analyses, I found out that I had incorrectly put the labels of significant differences in figure 4(c). This is now adjusted, but has no influence on the content or conclusions of the manuscript.

We have finalized the manuscript and formatted the figures according to the manuscript preparation guidelines. Please do not hesitate to contact me in case any issues appear.

Best regards,

Marijn Van de Broek